# Lidar-radar synergistic method to retrieve ice, supercooled and mixed-phase clouds properties

Clémantyne Aubry [1, 2], Delanoë Julien [1], Silke Groß [2], Florian Ewald [2], Frédéric Tridon [3], Olivier Jourdan [4], and Guillaume Mioche [4]

[1]Laboratoire Atmosphère, Milieux et Observations Spatiales, IPSL, UVSQ Université Paris-Saclay, Sorbonne Université, CNRS, Guyancourt, France
[2]Deutsches Zentrum für Luft- und Raumfahrt (DLR), Institut für Physik der Atmosphäre, Oberpfaffenhofen, Germany
[3]DIATI, Politecnico di Torino, Turin, Italy
[4]Laboratoire de Météorologie Physique, OPGC, Aubière, France

**Correspondence:** Clémantyne Aubry (clemantyne.aubry@latmos.ipsl.fr)

**Abstract.**

Mixed-phase clouds are not well represented in climate and weather forecasting models, due to a lack of the key processes controlling their life cycle. Developing methods to study these clouds is therefore essential, despite the complexity of mixed-phase cloud processes and the difficulty of observing two cloud phases simultaneously. We propose in this paper a new method to retrieve the microphysical properties of mixed-phase clouds, ice clouds and supercooled water clouds using airborne or satellite radar and lidar measurements, called VarPy-mix. This new approach extends an existing variational method developed for ice clouds retrieval using lidar, radar and passive radiometers. We assume that the attenuated lidar backscatter $\beta$ at $532\,\mathrm{nm}$ is more sensitive to particle concentration and is consequently mainly sensitive to the presence of supercooled water. In addition, radar reflectivity $Z$ at $95\,\mathrm{GHz}$ is sensitive to the size of hydrometeors and hence more sensitive to the presence of ice particles. Consequently, in the mixed-phase the supercooled droplets are retrieved with the lidar signal and the ice particles with the radar signal, meaning that the retrieval rely strongly on *a priori* and errors values. This method retrieves then simultaneously the visible extinction for ice $\alpha_{\mathrm{ice}}$ and liquid $\alpha_{\mathrm{liq}}$ particles, the ice and liquid water contents IWC and LWC, the effective radius of ice $r_{e,\mathrm{ice}}$ and liquid $r_{e,\mathrm{liq}}$ particles and the ice and liquid number concentrations $N_{\mathrm{ice}}$ and $N_{\mathrm{liq}}$. Moreover, total extinction $\alpha_{\mathrm{tot}}$, total water content TWC and total number concentration $N_{\mathrm{tot}}$ can also be estimated. As the retrieval of ice and liquid is different, it is necessary to correctly identify each phase of the cloud. To this end, a cloud phase classification is used as input to the algorithm and has been adapted for mixed-phase retrieval. The data used in this study are from DARDAR-MASK v2.23 products, based on the CALIOP lidar and CPR radar observations, respectively from the CALIPSO and CloudSat satellites belonging to the A-Train constellation launched in 2006. Airborne *in situ* measurements performed on the 7[th] April 2007 during the ASTAR campaign and collected under the track of CloudSat-CALIPSO are compared to the retrievals of the new algorithm to validate its performance. Visible extinctions, water contents, effective radii and number concentrations derived from *in situ* measurements and the retrievals showed similar trends and are globally in good agreement. The mean percent error between the retrievals and *in situ* is $39\,\%$ for $\alpha_{\mathrm{liq}}$, $398\,\%$ for $\alpha_{\mathrm{ice}}$, $49\,\%$ for LWC and $75\,\%$ for IWC. It is also important to note

that temporal and spatial collocations are not perfect, with a maximum spatial shift of 1.68 km and a maximum temporal shift about ten minutes between the two platforms. In addition, the sensitivity of remote sensing and *in situ* are not the same and *in situ* measurements uncertainties are between 25 % and 60 %.

## 1 Introduction

The current situation concerning climate change strongly impacts our society (IPCC, 2022), which leads to an interest in climate and weather forecasting. Clouds cover about 67 % of the Earth's atmosphere (King et al., 2013) and take an important part in Earth's water cycle and its radiation budget (Stephens, 2005). However, climate and weather prediction models still have a lack of knowledge in some situations and scenarios where clouds, especially mixed-phase clouds, remain one of the main sources of uncertainty, due to the complexity of the related processes. Mixed-phase clouds occur at all latitudes and more significantly at mid- and high-latitudes (Choi et al., 2010; Shupe, 2011) and are a coexisting mixture of three phases of water: ice particles, supercooled droplets and water vapor at temperatures between $-40\,°C$ and $0\,°C$. This coexistence implies complex formation processes, such as primary ice nucleation (Meyers et al., 1992), secondary ice production (Field et al., 2017; Kanji et al., 2017) and ice deposition (Meyers et al., 1992), and growing processes, such as the Wegener-Bergeron-Findeisen process (Wegener, 1911; Bergeron, 1935; Findeisen, 1938), water vapor deposition (Song and Lamb, 1994), aggregation (Hobbs et al., 1974) and riming (Hallett and Mossop, 1974). Since liquid and ice particles influence the shortwave and longwave radiation differently (Matus and L'Ecuyer, 2017), the fraction of liquid and ice particles significantly affects the radiative properties of mixed-phase clouds, altering the radiative balance of the Earth's atmospheric system. Moreover, all these processes are difficult to represent in numerical model (Morrison et al., 2008, 2012) and mixed-phase clouds that are not well represented in models can introduce significant biases, such as a misrepresentation of the cloudy state (Pithan et al., 2014). For that reason, it is crucial to have information on mixed-phase clouds microphysics in order to reduce the uncertainties in climate and weather prediction.

The localization and lifetime of the mixed-phase in a cloud differ according to the type of clouds and can make their observation challenging. The difference of water vapor saturation over ice and liquid makes the mixed-phase condensationally unstable and only exists for a limited time (Korolev et al., 2017). One way of observing these clouds is to use active remote sensing instruments. They can be onboard an aircraft or a satellite allowing to probe clouds on a large scale with vertical profiles seen from above. They are then useful to detect the mixed-phase layer at cloud top, which is typically the case in arctic boundary layer clouds (Gayet et al., 2009; Mioche et al., 2017). Each instrument has its own characteristics and a specific sensitivity that depends notably on the instrument wavelength. On one hand, the lidar measures the attenuated backscatter $\beta$ $[\mathrm{m^{-1}.sr^{-1}}]$, which corresponds to the energy backscattered by the targets and is affected by the atmospheric transmission. At a wavelength between 355 nm and 1064 nm, the lidar attenuated backscatter is more sensitive to the concentration of hydrometeors and can detect small cloud particles and aerosols. However, this signal can be attenuated or extinguished by a region with high particle concentration and cannot give information below this cloud layer. On the other hand, the radar measures the reflectivity $Z$ $[\mathrm{mm^6.m^{-3}}]$ typically at 35 or 95 GHz for cloud radars. At these wavelength, the radar reflectivity is more sensitive to the particle size and the signal can penetrate thick clouds (Delanoë et al., 2013; Cazenave et al., 2019). Consequently, in mixed-

phase clouds the lidar is more sensitive to highly concentrated liquid droplets and gives a strong backscatter signal. On the other hand, the radar reflectivity of liquid droplets is weaker than ice particles. As a result, both instruments complement each other. These measurements can then be used in algorithms to retrieve microphysical cloud properties such as the visible extinction $\alpha$,

the ice and liquid water contents (IWC and LWC) and the total number concentration $N_{\text{tot}}$.

Lidar-radar synergistic methods were first proposed by Intrieri et al. (1993), Donovan and van Lammeren (2001), Tinel et al. (2005) and Mitrescu et al. (2005) to retrieve ice clouds properties where both instrument overlap. Algorithms as VarCloud (Delanoë and Hogan, 2008) and 2C-ICE (Deng et al., 2010) were later developed to retrieve ice clouds properties all along the instruments profile using the Cloud Profiling Radar (CPR) onboard CloudSat (Stephens et al., 2002), the Cloud-Aerosol Lidar

with Orthogonal Polarization (CALIOP) onboard CALIPSO (Winker et al., 2003) and additionally radiometric information for VarCloud. For the EarthCARE mission (Illingworth et al., 2015) the unified synergistic retrieval algorithm CAPTIVATE (Mason et al., 2022) uses the ATmospheric LIDar (ATLID), the Cloud Profiling Radar (CPR), and the MultiSpectral Imager (MSI) data to retrieve clouds, precipitations and aerosols properties.

The variational method VarCloud, developed by Delanoë and Hogan (2008), aims to retrieve ice clouds properties using

radar, lidar and radiometric data synergy. Since, this algorithm has been improved with new parameterization for ice clouds retrievals (Ceccaldi, 2014; Cazenave et al., 2019), allowing more flexibility. As a result, it can process data from different airborne or spaceborne instruments platforms. In the mixed-phase, the algorithm only retrieve ice properties with the radar signal. The algorithm current version is called in this paper VarPy-ice and is described in detail in Cazenave (2019) thesis, pages 107 to 113. Our method, VarPy-mix, aims to retrieve simultaneously ice, supercooled water and mixed-phase clouds

properties with lidar and radar synergy, based on VarPy-ice to retrieve ice clouds. Each cloud phase is not processed in the same way. The ice clouds are retrieved with both instruments while the mixed-phase retrieval is divided in two parts: the ice particles are retrieved with the radar signal and the supercooled water with the lidar signal. Besides, supercooled water clouds are retrieved with the lidar signal only. Therefore, the retrieval relies strongly on *a priori* and errors values. Additionally, this flexible algorithm can be apply on several radar-lidar platforms, airborne or spaceborne. As a starting point, these changes

were developed with CloudSat and CALIPSO instruments datasets. These data have a large, robust and proven classification algorithmic statistics as well as existing cases of collocation with *in situ* measurements.

In this paper, we first describe in Sect. 2 the general points of both version of VarPy before going into details of the new version structure. In addition, the processed cloud phases are presented in this section, along with an adaptation of the cloud phase classification dedicated to the mixed-phase and supercooled water. Next, Sect. 3 presents a case of mixed-phase at the

top of an ice cloud for which microphysical properties are retrieved using VarPy-mix and compared with *in situ* measurements. Finally, the last section is dedicated to a conclusion and to an outlook on future work.

## 2 Methodology

### 2.1 Variational method

#### 2.1.1 Description of VarPy

The radar reflectivity $Z$ [mm$^6$.m$^{-3}$] and the lidar attenuated backscatter $\beta$ [m$^{-1}$.sr$^{-1}$] are linked to the clouds microphysical vertical structure. For example the water content is strongly correlated with the reflectivity (Atlas, 1954) and the lidar backscatter is related to the cloud extinction $\alpha$. We can relate this situation to an inverse problem given by:

$$\mathbf{Y} = f(\mathbf{X}) + \epsilon \tag{1}$$

The vector $\mathbf{Y}$ is the observation vector composed of the measured radar reflectivity $Z_{\mathrm{obs}}$ and the lidar attenuated backscatter
$\beta_{\mathrm{obs}}$. The vector $\mathbf{X}$ is composed of the quantities that describe the system, e.g. some clouds microphysical properties. The function $f$ is the "forward function" (Rodgers, 2000, p. 14) and in our case represents the lidar and the radar forward models. These models and measurements are associated to specific uncertainties that can be presented by the error vector $\epsilon$. According to the values of the vector $\mathbf{X}$, called hereafter the state vector, the forward models predict reflectivity and backscatter values, respectively noted $Z_{\mathrm{fwd}}$ and $\beta_{\mathrm{fwd}}$. These forward modeled values are afterwards compared to the measurements. The difference
between $\mathbf{Y}$ and the predicted values is used to update the state vector via the Gauss-Newton method. New values of $Z_{\mathrm{fwd}}$ and $\beta_{\mathrm{fwd}}$ are computed with the forward models and Look Up Table (LUT, detailed in Sect. 2.1.2) until convergence occurs according to a $\chi^2$ test. The solution is defined by the state vector at the last iteration when the solution converges. These values are used in combination with a LUT to retrieve the desired microphysical properties. The diagram of Fig. 1 summarizes the whole structure of the variational scheme.

The two main inputs of VarPy are the observations vector $\mathbf{Y}$ (box 2 of Fig. 1, Eq. (2)) and the initialized state vector $\mathbf{X_0}$ (box 1 of Fig. 1), given by Eq. (5) with first guess values from Table 1, as explained in Sect. 2.1.2. The natural logarithm is applied to the variables of $\mathbf{X}$ and $\mathbf{Y}$ to avoid the unphysical possibility of retrieving negative values. Both vectors are defined for one measurement profile and as a function of the distance from the instrument. Radar and lidar do not have the same amount of values per profile (also called hereafter gate): there are $q$ values of $\ln(Z_{\mathrm{obs}})$ for a profile and $p$ values for $\ln(\beta_{\mathrm{obs}})$. Then the
observations vector $\mathbf{Y}$ is defined for a single profile as follows:

$$\mathbf{Y} = \begin{pmatrix} \ln(Z_{\mathrm{obs},0}) \\ \vdots \\ \ln(Z_{\mathrm{obs},q}) \\ \\ \ln(\beta_{\mathrm{obs},0}) \\ \vdots \\ \ln(\beta_{\mathrm{obs},p}) \end{pmatrix} \tag{2}$$

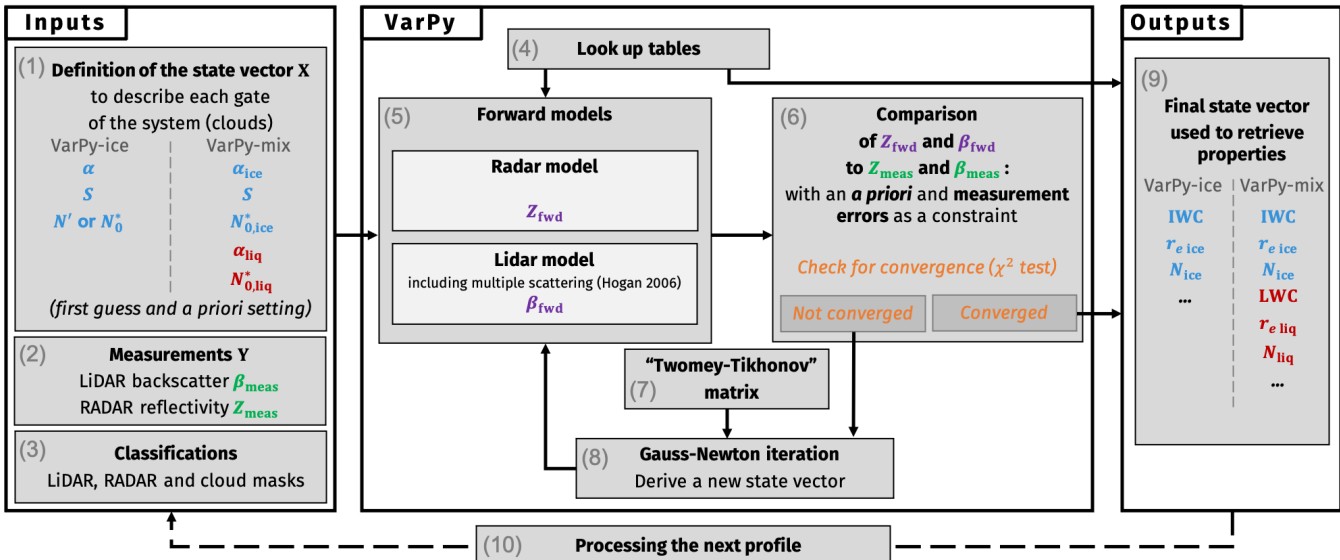

**Figure 1.** Diagram showing the sequence of operations executed by the VarPy retrieval method.

To retrieve ice properties, the state vector is composed of the visible extinction $\alpha$ [m$^{-1}$], the extinction-to-backscatter ratio $S$ [sr] and $N'$ which is related to the normalized number concentration parameter $N_0^*$ [m$^{-4}$] via the following relationship:

$$N' = \frac{N_0^*}{\alpha^\gamma} \tag{3}$$

where $\gamma$ is an empirically determined coefficient normalizing $N'$ (Delanoë and Hogan, 2010; Delanoë et al., 2014). Values for this coefficient are shown in Table 1. For $n$ measurements gates, the state vector is composed of $n$ values of $\ln(\alpha)$. However, $N'$ is not retrieved for each gate. A cubic-spline basis function interpolates the $N'$ profile with a number concentration parameter spacing factor $\eta_N$ set to 4 and decreases the number of $N'$ values to $m$ such that smooth variation in range is guaranteed (Hogan, 2007; Delanoë and Hogan, 2008). This improves computing efficiency by reducing calculation time.

The lidar ratio is assumed to be a function of temperature $T$ [°C], adapted from Platt et al. (2002) and derived using radar-lidar data from previous version of DARDAR (Cazenave et al., 2019). Consequently, the lidar ratio $S$ is not represented in the state vector for each gate but by the two coefficients $a_{\ln(S)}$ and $b_{\ln(S)}$ that are the slope and the intercept coefficient from the temperature dependence relationship (Eq. (4)). As a result, for this configuration of the state vector, the dimension of the lidar ratio $S$ is given by $k = 2$. For VarPy-ice, the average retrieved lidar ratio equals $35 \pm 10$ sr for a temperature range from $-60$
125   °C to $-20$ °C (Cazenave et al., 2019).

$$\ln(S) = a_{\ln(S)} + b_{\ln(S)} \cdot T \tag{4}$$

Thereby for VarPy-ice, the state vector to retrieve a profile of $n$ gates is as following:

$$\mathbf{X}_{\text{ice}} = \begin{pmatrix} \ln(N'_0) \\ \vdots \\ \ln(N'_m) \\ \\ a_{\ln(S)} \\ b_{\ln(S)} \\ \\ \ln(\alpha_0) \\ \vdots \\ \ln(\alpha_n) \end{pmatrix} \tag{5}$$

The update of the state vector (box 8 of Fig. 1) is given by:

$$\mathbf{X_{k+1}} = \mathbf{X_k} + \mathbf{H}^{-1} \cdot (\mathbf{J}^{\mathsf{T}} \cdot \mathbf{R}^{-1} \cdot (\mathbf{Y}_{\text{osb}} - \mathbf{Y}_{\text{fwd}}) - \mathbf{B}^{-1} \cdot (\mathbf{X}_k - \mathbf{X}_a) - \mathbf{T} \cdot \mathbf{X}_k) \tag{6}$$

with $\mathbf{J}$ the Jacobian matrix that contains the partial derivative of $\ln(Z_{\text{fwd}})$ and $\ln(\beta_{\text{fwd}})$ with respect to each element of the state vector (box 5 of Fig. 1), $\mathbf{H}$ is the Hessian matrix given by Eq. (7), $\mathbf{R}$ is the error covariance matrix of the observations, $\mathbf{B}$ is the error covariance matrix of the *a priori* (explained in Sect. 2.2.1), and $\mathbf{T}$ is the "Twomey-Tikhonov" matrix (box 7 of Fig. 1; Rodgers, 2000) used to smooth the extinction profile.

$$\mathbf{H} = \mathbf{J}^{\mathsf{T}} \cdot \mathbf{R}^{-1} \cdot \mathbf{J} + \mathbf{B}^{-1} + \mathbf{T} \tag{7}$$

Each measurement is limited by the instrument performance and the signal-to-noise ratio. This is notably the case for the lidar and this can affect the retrieval of the extinction (Hogan et al., 2006). To limit the impact of measurement noise, a "Twomey-Tikhonov" matrix $\mathbf{T}$ can be used to penalize the second derivative of the state vector variables profile, especially the extinction. $\mathbf{T}$ is a square matrix and is defined, at dimension 6, by:

$$\mathbf{T} = \kappa \times \begin{pmatrix} 1 & -2 & 1 & 0 & 0 & 0 \\ -2 & 5 & -4 & 1 & 0 & 0 \\ 1 & -4 & 6 & -4 & 1 & 0 \\ 0 & 1 & -4 & 6 & -4 & 1 \\ 0 & 0 & 1 & -4 & 5 & -2 \\ 0 & 0 & 0 & 1 & -2 & 1 \end{pmatrix} \tag{8}$$

where $\kappa$ is a coefficient that sets the smoothness degree of $\mathbf{T}$. The dimensions of the final matrix $\mathbf{T}$ used by the algorithm correspond to those of the state vector depending on the version of VarPy. As we only want to smooth the extinction profile, the values of $\mathbf{T}$ corresponding to the lidar ratio $S$ and the number concentration parameter $N'$ are set to 0.

The Jacobian is a product of the forward models (box 5 of Fig. 1) and its composition depends on the structure of the state vector. For VarPy-ice, this matrix is given by Eq. (9) with a dimension of $(p+q) \times (m+2+n)$:

$$\mathbf{J} = \begin{pmatrix} \dfrac{\partial \beta_0}{\partial N_0'} & \cdots & \dfrac{\partial \beta_0}{\partial N_m'} & \dfrac{\partial \beta_0}{\partial a_{\ln(S)}} & \dfrac{\partial \beta_0}{\partial b_{\ln(S)}} & \dfrac{\partial \beta_0}{\partial \alpha_0} & \cdots & \dfrac{\partial \beta_0}{\partial \alpha_n} \\ \vdots & \ddots & \vdots & \vdots & \vdots & \vdots & \ddots & \vdots \\ \dfrac{\partial \beta_p}{\partial N_0'} & \cdots & \dfrac{\partial \beta_p}{\partial N_m'} & \dfrac{\partial \beta_p}{\partial a_{\ln(S)}} & \dfrac{\partial \beta_p}{\partial b_{\ln(S)}} & \dfrac{\partial \beta_p}{\partial \alpha_0} & \cdots & \dfrac{\partial \beta_p}{\partial \alpha_n} \\ \dfrac{\partial Z_0}{\partial N_0'} & \cdots & \dfrac{\partial Z_0}{\partial N_m'} & \dfrac{\partial Z_0}{\partial a_{\ln(S)}} & \dfrac{\partial Z_0}{\partial b_{\ln(S)}} & \dfrac{\partial Z_0}{\partial \alpha_0} & \cdots & \dfrac{\partial Z_0}{\partial \alpha_n} \\ \vdots & \ddots & \vdots & \vdots & \vdots & \vdots & \ddots & \vdots \\ \dfrac{\partial Z_q}{\partial N_0'} & \cdots & \dfrac{\partial Z_q}{\partial N_m'} & \dfrac{\partial Z_q}{\partial a_{\ln(S)}} & \dfrac{\partial Z_q}{\partial b_{\ln(S)}} & \dfrac{\partial Z_q}{\partial \alpha_0} & \cdots & \dfrac{\partial Z_q}{\partial \alpha_n} \end{pmatrix} \tag{9}$$

For better readability, the indices $_{\mathrm{fwd}}$ of $Z$ and $\beta$ are not displayed and the natural logarithm of $Z$, $\beta$, $N_0^*$ and $\alpha$ are not written.

### 2.1.2 State vector parameterization

During the iterative process, the state vector variables are used by the forward models (radar and lidar) to compute the radar reflectivity $Z_{\mathrm{fwd}}$ and the lidar backscatter $\beta_{\mathrm{fwd}}$. The lidar forward model differs from the radar forward model because an additional step is required to obtain $\beta_{\mathrm{fwd}}$ with the equivalent area radius $r_a$ and the multiscatter code from Hogan (2006) (box 5 of Fig. 1). To obtain $Z_{\mathrm{fwd}}$ and $r_a$, the ratio $\dfrac{\alpha}{N_0^*}$ derived from the state vector is linked to these variables via an one-dimensional LUT (box 4 of Fig. 1), which is also used to retrieve the cloud microphysical properties (box 9 of Fig. 1) as the effective radius $r_e$ and the ice water content IWC. The ice clouds properties can be retrieved with two types of LUT. The "Heymsfield Composite" (HC) LUT uses the Transition Matrix Method (T-matrix) and the mass-size relationship from Heymsfield et al. (2010). The "Brown and Francis modified" (BF) LUT is based on a combination of Brown and Francis (1995) and Mitchell (1996) mass-size relationships. These LUT are used for DARDAR-CLOUD v3.00 and v3.10 products (Delanoë, 2023a, b) and more details about them can be found in Delanoë et al. (2014) and Cazenave (2019). For both VarPy-ice and -mix, both LUT can be used to retrieve the ice properties and one must be selected beforehand. Regarding the retrieval of liquid part of the mixed-phase and supercooled water clouds, a LUT has been created and more details can be found in Sect. 2.2.2.

The LUT setting also involves defining the *a priori* and first guess values of the state vector. The first guess values are used to initialize the state vector for the forward models before the first iteration, corresponding to $\mathbf{X_0}$. The *a priori* values are important for regions where only one instrument is available and this constrains the scheme towards temperature dependent empirical relationships. We have postulated in the Sect. 2.1.1 that the lidar ratio is given by a temperature-dependent relationship (Eq. (4)) and *a priori* and first guess values are listed in Table 1. For the number concentration parameter $\ln(N')$, the *a priori* and first guess values are also given as a function of the temperature $T$:

$$\ln(N') = (A + B \cdot T) \tag{10}$$

**Table 1.** *A priori* and first guess values for each variable of the state vector.

| Variables | Values |
|-----------|--------|
| $a_{\ln(S)}$ | 3.18 |
| $b_{\ln(S)}$ | $-0.0086$ |
| $A_{\mathrm{BF}}$ | 22.234435 |
| $B_{\mathrm{BF}}$ | $-0.090736$ |
| $\gamma_{\mathrm{BF}}$ | 0.61 |
| $A_{\mathrm{HC}}$ | 21.94 |
| $B_{\mathrm{HC}}$ | $-0.095$ |
| $\gamma_{\mathrm{HC}}$ | 0.67 |
| $\ln(\alpha_{\mathrm{ice}})$ | $-7$ |
| $\ln(N^*_{0,\mathrm{liq}})$ | 30 |
| $\ln(\alpha_{\mathrm{liq}})$ | $-5$ |

Table 1 lists the values of $A$ and $B$ used for each mass-size relationship (BF and HC). The coefficient $\gamma$ linking $N^*_0$ to $\alpha$ and $N'$ differs according to the mass-size relationship and the values are also given in Table 1. The *a priori* and first guess values for the extinction are constant values.

### 2.1.3 Definition of VarPy versions

Before going into details of the adaptations made in VarPy-mix to retrieve supercooled water and mixed-phase clouds, we describe in this section the main assumptions on the instruments used for VarPy-ice and -mix.

VarPy-ice retrieves ice properties from radar and lidar measurements, including ice from mixed-phase layers. Since the lidar signal is more sensitive to liquid droplets than ice particles, it cannot be used in VarPy-ice to retrieve ice properties of mixed-phase. Therefore, every lidar gate below the mixed-phase layer cannot be used due to the attenuation of the liquid droplets in the lidar signal. Consequently, the mixed-phase and the ice cloud below are not retrieved via radar-lidar synergy but only with the radar signal and the state vector *a priori* values.

The main hypothesis for the VarPy-mix version is to consider the ice and liquid parts of the mixed-phase separately and retrieve the liquid part with the lidar signal and the ice part with the radar signal. This hypothesis is based on the sensitivity of the instruments, explained in the introduction (Sect. 1). The aim of this version of the algorithm is to be able to retrieve several cloud phases using the same variational method, but with a structure and parameterization that are adapted to supercooled water and the mixed-phase. A large part of VarPy-ice has been preserved to maintain the strengths of the method and the consistency of the results.

## 2.2 New configuration of the state vector to retrieve ice and supercooled water simultaneously

For the new version of the algorithm, the state vector needs to be adapted to also retrieve supercooled water properties. The special case of the mixed-phase has to be taken into account. The supercooled water and the ice particles properties are retrieved separately for mixed-phase. The state vector is consequently divided in two parts: one part of the variables retrieves ice properties and the other part retrieves liquid properties. The ice particles of the mixed-phase are included in the ice part and the supercooled droplets are in the liquid part. The composition of the state vector differs from the previous version and will be described in the following paragraphs.

As the liquid droplet concentration does not depend on the air temperature like for ice particles, the temperature-dependent concentration parameter $N'$ is not required to retrieve liquid cloud properties. For this, we decided to use $N_0^*$ in the state vector, instead of $N'$. It can be noted that VarPy-ice algorithm has also the possibility to retrieve ice properties using the normalized number concentration parameter $N_0^*$. This enables the VarPy-mix ice retrieval to be compared with VarPy-ice retrieval to avoid any inconsistencies. We include this variable for each state vector part, so there is $N_{0,\text{ice}}^*$ for the ice part and $N_{0,\text{liq}}^*$ for the liquid part. Choosing $N_0^*$ allows to keep the *a priori* and first guess values for the ice with the following temperature dependent relationship:

$$\ln(N_{0,\text{ice}}^*) = (A + B \cdot T) + \gamma \cdot \ln \alpha_{\text{ice}} \tag{11}$$

This relation is based on the Eq. (10) to calculate $N'$ *a priori* and first guess values. To keep the old scheme benefits, the cubic-spline basis function interpolates the $N_{0,\text{ice}}^*$ values with a spacing factor $\eta_N$ set to $4$. It is unusable for the liquid group since supercooled layer are thin and corresponds to too few gates.

The extinction $\alpha$ is still part of the state vector. Like for $N_{0,\text{ice}}^*$ and $N_{0,\text{liq}}^*$, the extinction is divided into two variable: $\alpha_{\text{ice}}$ for the ice properties and $\alpha_{\text{liq}}$ for the liquid ones. Both are defined for each gate of a profile. Regarding the lidar ratio $S$, we keep the same configuration in the state vector with the two coefficients $a_{\ln(S)}$ and $b_{\ln(S)}$. Table 2 lists the value of the lidar ratio at different wavelength and according to particle size or type. As a result, we make the assumption that the lidar ratio is constant for liquid droplets (Pinnick et al., 1983). Consequently, the lidar ratio is defined only for ice gates in the state vector and its value is fixed at $18.6$ sr for supercooled water (pure or in mixed-phase) at $532$ nm.

For $n_i$ the number of ice gates, $n_l$ the number of liquid gates and $m_i$ defined in the same way as $m$ depending on the spacing parameter $\eta_N$, we end up with the new state vector given by Eq. (12):

**Table 2.** Lidar ratio $S$ for liquid droplet depending on cloud type, particle size and lidar wavelength.

| Source | Particle or cloud type | Wavelength $\lambda$ [nm] | $S$ [sr] |
|---|---|---|---|
| Pinnick et al. (1983) | Spherical water droplets | 1064 | 18.2 |
| | | 632 | 17.7 |
| O'Connor et al. (2004) | Median equivolumetric diameter between 8 and 20 μm | 905 | $18.8 \pm 0.8$ |
| | | 532 | $18.6 \pm 1.0$ |
| | | 355 | $18.9 \pm 0.4$ |
| Hogan et al. (2003) | Mie theory and distributions with median volume diameters between 5 and 50 μm | 905 | 18.75 |

$$\mathbf{X}_{\text{mix}} = \begin{pmatrix} \ln(N^*_{0,\text{ice},0}) \\ \vdots \\ \ln(N^*_{0,\text{ice},m_i}) \\ a_{\ln(S)} \\ b_{\ln(S)} \\ \ln(\alpha_{\text{ice},0}) \\ \vdots \\ \ln(\alpha_{\text{ice},n_i}) \\ \\ \ln(N^*_{0,\text{liq},0}) \\ \vdots \\ \ln(N^*_{0,\text{liq},n_l}) \\ \ln(\alpha_{\text{liq},0}) \\ \vdots \\ \ln(\alpha_{\text{liq},n_l}) \end{pmatrix} \qquad (12)$$

For VarPy-ice, a single "Twomey-Tikhonov" matrix of dimension $(m+k+n) \times (m+k+n)$ is applied for the entire extinction profile. However, the extinction values of liquid droplets is different from ice particles, and it is therefore unsuitable to use a single "Twomey-Tikhonov" matrix on a profile with simultaneously ice and supercooled water. As we want to smooth the
215 extinction profile for both ice and liquid parts, we decided to smooth them out separately and not to use a single "Twomey-Tikhonov" matrix to smooth an entire profile. Consequently, for VarPy-mix, a method has been developed to separate different sections of a profile according to the smoothing to be applied. The separation is made between ice, liquid and where there is clear sky. Then, one "Twomey-Tikhonov" matrix is applied to each section. The dimension of the final matrix is $(m_i+k+n_i+$

$2 \times n_l) \times (m_i + k + n_i + 2 \times n_l)$. The smoothness coefficient $\kappa$ is set to 100 for VarPy-ice and this parameterization is kept for the ice part of VarPy-mix. Whereas, the coefficient applied to the liquid water is different and set to 10, since the thickness of the detected liquid layer is smaller than ice layer.

### 2.2.1 *A priori* error covariance matrix

Generally, radar and lidar signals do not both cover simultaneously the entire vertical cloud profile. In many cases of ice clouds, lidar in downward direction first detects the top of the cloud, while radar only detects deeper cloud regions down to the ground. The lidar signal will not detect the lower layers of the cloud if it gets attenuated or extinguished. To ensure that the results tend towards physical values in regions where a single instrument is available, state vector *a priori* parameterization and errors are used. The *a priori* errors are defined by the *a priori* error covariance matrix $\mathbf{B}$ and express how strong is the constrain of the *a priori*. This matrix is composed of the error variances of the state vector *a priori* $\sigma^2$. In the simplest case where no information propagates between gates, this matrix is diagonal.

To overcome the limitation of single instrument retrieval, the matrix $\mathbf{B}$ can be used to spread information in height. Additional off-diagonal elements can be added to propagate information from synergistic regions to single instrument ones. In VarPy-ice (Hogan, 2007; Delanoë and Hogan, 2008), the off-diagonal terms of $\mathbf{B}$ corresponding to $N'$ are given by:

$$B_{i,j} = B_{i,i} \times e^{-\dfrac{|z_j - z_i|}{z_0}} \tag{13}$$

where $z_0$ is the decorrelation distance, a parameter set to $600$ m for VarPy (initially set to $1$ km for VarCloud). This value is set for CloudSat-CALIPSO and can be adapated to the resolution of the data used.

In VarPy-mix version, the structure of $\mathbf{B}$ has been adapted to the composition of the new state vector. In order to keep the same configuration as VarPy-ice, the off-diagonal terms are calculated for $N^*_{0,\text{ice}}$ only. As a result, $\mathbf{B}$ remains diagonal regarding the other variables. The *a priori* error variances values for both VarPy-ice and -mix are listed in Table 3 and are assumed to be constant with height.

Besides, the dimensions of the matrices $\mathbf{U}$ and $\mathbf{M}$, used for the calculation of the error covariance matrix of the state vector $\mathbf{S_x}$ (refer to Appendix A of Delanoë and Hogan 2008 for more information), have been adapted to the number of variables in $\mathbf{X}_{\text{mix}}$ and their dimension.

### 2.2.2 Normalized Droplet Size Distribution for liquid Look Up Table

For VarPy-ice and VarPy-mix, ice properties are retrieved using dedicated LUTs. These are created using the particle size distribution of ice particles (Delanoë et al., 2014). However, the particle size distribution differs between ice particles and liquid droplets, meaning that LUTs dedicated to the retrieval of ice properties cannot be used to retrieve liquid properties. The solution is to define a Droplet Size Distribution (DSD) for liquid droplets to create a LUT dedicated to liquid properties retrieval. Regarding literature, there are two types of distribution: the gamma distribution (Miles et al., 2000) and the log-

**Table 3.** *A priori* error variances used in VarPy for the *a priori* error covariance matrix $\mathbf{B}$

| Variables | Values |
|:---:|:---:|
| $\sigma_{\ln(N')}$ | 1 |
| $\sigma_{\ln(N_{0,\text{ice}}^*)}$ | 1 |
| $\sigma_{a_{\ln(S)}}$ | 0.1 |
| $\sigma_{b_{\ln(S)}}$ | 0.0001 |
| $\sigma_{\ln(\alpha_{\text{ice}})}$ | 5 |
| $\sigma_{\ln(N_{0,\text{liq}}^*)}$ | 1 |
| $\sigma_{\ln(\alpha_{\text{liq}})}$ | 5 |

normal distribution (Frisch et al., 1995; Fielding et al., 2015). For this study we use the following log-normal relationship

defined by Frisch et al. (1995):

$$n(r) = \frac{N_{\text{liq}}}{\sigma\sqrt{2\pi}} e^{-\frac{(\ln(r) - \ln(r_0))^2}{2\sigma^2}} \tag{14}$$

where $n(r)$ is the number concentration at a given cloud droplet radius $r$ [μm], $N_{\text{liq}}$ is the total number of liquid droplets per unit volume [m$^{-3}$], $r_0$ is the modal radius [μm], and $\sigma$ is the geometric standard deviation. The k$^{\text{th}}$ moment $\langle r^k \rangle$ of this distribution can be expressed as following:

$$\langle r^k \rangle = \frac{1}{N_{\text{liq}}} \int_0^\infty n(r) r^k \, dr \tag{15}$$

It permits to relate the following variables to $D_m$ [m] (proportional to the ratio between the fourth moment and the the third moment):

- – the reflectivity $Z$ [mm$^6$.m$^{-3}$], which is proportional to the sixth moment of the DSD

- – the extinction $\alpha$ [m$^{-1}$], which is proportional to the second moment

– the liquid water content LWC [kg.m$^{-3}$], which is proportional to the third moment

- – the effective radius $r_e$ [m], which is proportional to the ratio between the third moment and the the second moment

- – the equivalent area radius $r_a$ [m], which is equivalent to $r_e$ for droplets

- – the total number of liquid droplets per unit volume $N_{\text{liq}}$ [m$^{-3}$]

Those quantities are then normalized by $N_0^*$ [m$^{-4}$] which can also be expressed as a function of $D_m$ using the moments of the

distribution (proportional to the ratio between the third moment to the fifth power and the fourth moment to the fourth power).

The LUT ends up to be composed of $\frac{Z}{N_0^*}, \frac{\alpha}{N_0^*}, \frac{LWC}{N_0^*}, \frac{N_{\text{liq}}}{N_0^*}$, $r_a$ and $r_e$ as a function of $D_m$. As with ice LUT, the liquid one is used in two steps of the algorithm with the ratio $\frac{\ln(\alpha_{\text{liq}})}{\ln(N_{0,\text{liq}}^*)}$ from the state vector values. This ratio is used to retrieve the corresponding value in LUT, by interpolation. First, at each iteration, to predict $\ln Z_{\text{fwd}}$, $\ln \beta_{\text{fwd}}$ (via $\ln r_a$ and the fast multiple-scattering model of Hogan 2006) and the Jacobian terms with the forward models. Then with the final state vector, the ratio $\frac{\ln(\alpha_{\text{liq}})}{\ln(N_{0,\text{liq}}^*)}$ permits to obtain LWC, $r_{e,\text{liq}}$ and $N_{\text{liq}}$ (box 9 of Fig. 1).

As explained in Sect. 2.1.1, two LUTs are available to retrieve ice properties. They are both implemented in VarPy-mix to retrieve the ice part of the mixed-phase. Besides, they are defined in terms of the mean volume-weighted melted-equivalent diameter, which makes them very similar from the liquid LUT for small radius. This ensures scientific consistency and algorithm flexibility.

### 2.2.3 Jacobians

The Jacobian depends on the state vector composition and is different between VarPy-ice and VarPy-mix. The structure of the Jacobian $\mathbf{J}$ for VarPy-mix is shown by Eq. (16).

$$
\mathbf{J} = \begin{pmatrix}
\frac{\partial \beta_0}{\partial N_{0,i,0}^*} & \cdots & \frac{\partial \beta_0}{\partial N_{0,i,m_i}^*} & \frac{\partial \beta_0}{\partial a_{\ln(S)}} & \frac{\partial \beta_0}{\partial b_{\ln(S)}} & \frac{\partial \beta_0}{\partial \alpha_{i,0}} & \cdots & \frac{\partial \beta_0}{\partial \alpha_{i,n_i}} & \frac{\partial \beta_0}{\partial N_{0,l,0}^*} & \cdots & \frac{\partial \beta_0}{\partial N_{0,l,n_l}^*} & \frac{\partial \beta_0}{\partial \alpha_{l,0}} & \cdots & \frac{\partial \beta_0}{\partial \alpha_{l,n_l}} \\
\vdots & \ddots & \vdots & \vdots & \vdots & \vdots & \ddots & \vdots & \vdots & \ddots & \vdots & \vdots & \ddots & \vdots \\
\frac{\partial \beta_p}{\partial N_{0,i,0}^*} & \cdots & \frac{\partial \beta_p}{\partial N_{0,i,m_i}^*} & \frac{\partial \beta_p}{\partial a_{\ln(S)}} & \frac{\partial \beta_p}{\partial b_{\ln(S)}} & \frac{\partial \beta_p}{\partial \alpha_{i,0}} & \cdots & \frac{\partial \beta_p}{\partial \alpha_{i,n_i}} & \frac{\partial \beta_p}{\partial N_{0,l,0}^*} & \cdots & \frac{\partial \beta_p}{\partial N_{0,l,n_l}^*} & \frac{\partial \beta_p}{\partial \alpha_{l,0}} & \cdots & \frac{\partial \beta_p}{\partial \alpha_{l,n_l}} \\
\frac{\partial Z_0}{\partial N_{0,i,0}^*} & \cdots & \frac{\partial Z_0}{\partial N_{0,i,m_i}^*} & \frac{\partial Z_0}{\partial a_{\ln(S)}} & \frac{\partial Z_0}{\partial b_{\ln(S)}} & \frac{\partial Z_0}{\partial \alpha_{i,0}} & \cdots & \frac{\partial Z_0}{\partial \alpha_{i,n_i}} & \frac{\partial Z_0}{\partial N_{0,l,0}^*} & \cdots & \frac{\partial Z_0}{\partial N_{0,l,n_l}^*} & \frac{\partial Z_0}{\partial \alpha_{l,0}} & \cdots & \frac{\partial Z_0}{\partial \alpha_{l,n_l}} \\
\vdots & \ddots & \vdots & \vdots & \vdots & \vdots & \ddots & \vdots & \vdots & \ddots & \vdots & \vdots & \ddots & \vdots \\
\frac{\partial Z_q}{\partial N_{0,i,0}^*} & \cdots & \frac{\partial Z_q}{\partial N_{0,i,m_i}^*} & \frac{\partial Z_q}{\partial a_{\ln(S)}} & \frac{\partial Z_q}{\partial b_{\ln(S)}} & \frac{\partial Z_q}{\partial \alpha_{i,0}} & \cdots & \frac{\partial Z_q}{\partial \alpha_{i,n_i}} & \frac{\partial Z_q}{\partial N_{0,l,0}^*} & \cdots & \frac{\partial Z_q}{\partial N_{0,l,n_l}^*} & \frac{\partial Z_q}{\partial \alpha_{l,0}} & \cdots & \frac{\partial Z_q}{\partial \alpha_{l,n_l}}
\end{pmatrix}
\tag{16}
$$

For better readability, the indices $_{\text{fwd}}$ of $Z$ and $\beta$ are not displayed, the $_{\text{ice}}$ and $_{\text{liq}}$ indices of $N_0^*$ and $\alpha$ are replaced respectively by $_i$ and $_l$ indices and the natural logarithm of $Z$, $\beta$, $N_0^*$ and $\alpha$ are omitted. As for the state vector, we can divide the Jacobian in two parts: the derivatives of $\ln Z$ and $\ln \beta$ with respect to $\ln N_{0,\text{ice}}^*$, $a_{\ln(S)}$, $b_{\ln(S)}$ and $\ln \alpha_{\text{ice}}$ for the ice part (blue background color on Eq. (16)) and the derivatives of $\ln Z$ and $\ln \beta$ with respect to $\ln N_{0,\text{liq}}^*$ and $\ln \alpha_{\text{liq}}$ for the liquid part (red background color). For the mixed-phase both liquid and ice parts are used. However, each part is retrieved with only one instrument. Indeed, the radar is not used to retrieve the supercooled water neither in pure liquid clouds nor in mixed-phase clouds, therefore $\frac{\partial \ln(Z_j)}{\partial \ln(N_{0,\text{liq},k}^*)}$ and $\frac{\partial \ln(Z_j)}{\partial \ln(\alpha_{\text{liq},k})}$ are zero for any $j$ and $k$. The lidar is used to retrieve ice clouds properties but not the ice part of the mixed-phase. Then, $\frac{\partial \ln(\beta_j)}{\partial \ln(N_{0,\text{ice},k}^*)}$ and $\frac{\partial \ln(\beta_j)}{\partial \ln(\alpha_{\text{ice},k})}$ are zero for any $j$ and $k$ corresponding to mixed-phase gates.

### 2.3 Cloud phase classification

Ice particles and liquid droplets are processed differently, meaning that the hydrometeors identification is an important algorithm input and more significantly regarding the mixed-phase. The retrieval of clouds properties requires to distinguish the

different hydrometeors detected by the instruments - here the radar and the lidar. Therefore, according to the sensitivity of each instrument, a hydrometeor classification is established for each instrument. Lidar classification distinguishes aerosols and cloud phases, while radar classification identifies precipitations and clouds. Consequently, combining the lidar and radar classifications results in a more detailed cloud phase classification. These three classifications are additional inputs to the algorithm (box 3 of Fig. 1).

DARDAR-MASK v2.23 (Delanoë and Hogan, 2010) is a target categorization made by the combination of the 2B-GEOPROF CloudSat radar mask, the CALIPSO vertical lidar feature mask CAL-LID-L2-VFM and CALIPSO L1 measurements with a multi-threshold decision tree (Ceccaldi et al., 2013; Cazenave et al., 2019). VarPy algorithms use it in order to select the gates to process and how to process them. Table 4 shows the eighteen classes of the DARDAR-MASK v2.23 classification. The classes are not all processed. Currently, the algorithm process the "ice cloud", the "spherical or 2D ice", the "supercooled water", the "supercooled water and ice", the "highly concentrated ice particles", the "top of the convective tower" and the "multiple scattering due to supercooled water" classes. They are highlighted in bold in the Table 4. For VarPy-ice, these classes form a single group to be processed. On the other hand, two groups of classes have been defined for VarPy-mix. Table 5 presents the composition of these groups. The group called "ice" is composed of the following classes: "ice clouds", "spherical or 2D ice", "supercooled water and ice", "highly concentrated ice particles" and "top of convective towers". The "supercooled water", "supercooled water and ice" and "multiple scattering" classes define the "liquid" group. This distinction is necessary to process the different phases of the clouds separately. Nevertheless the "supercooled water and ice" class, called hereafter "mixed-phase", has the particularity to be processed in both the ice and liquid groups. In the current versions of VarPy, an intermediate classification is created with these groups, called the processed cloud phase classification. For the case used for this paper, the processed cloud phase classification is presented in Sect. 3 in Fig. 2 (c).

Supercooled water layers are detected and identified using the lidar signal. In order to distinguish between classes "supercooled water" an "supercooled water and ice", the radar signal is used. On the one hand, if the radar detects ice, the cloud phase classification identifies the area as "supercooled water and ice". On the other hand, where the radar does not detect particles (no radar signal) and the lidar backscatter is strong, it is categorized as "supercooled water" and the following gates are usually "multiple scattering due to supercooled water". For these gates, retrievals are based only on lidar measurements and *a priori* values.

To minimize misclassification, some adaptations of the cloud phase classification have been implemented. The first step is to avoid isolated gates that bias the retrieval. A method has been created to erode isolated supercooled water and mixed-phase gates. For supercooled water and multiple scattering phases, these gates are replaced by clear sky in the cloud phase classification and the same correction is made for the lidar classification. On the other hand for the mixed-phase, only the cloud phase classification is modified and the gates are replaced by ice gates. Afterwards, the next step is to correct some misclassification of the mixed-phase. A strong lidar backscatter signal ($\beta_{532} > 2.10^{-5}$ m$^{-1}$.sr$^{-1}$; Delanoë and Hogan, 2010) can be a detection of warm water, top of convective tower, highly concentrated ice particles or supercooled water. For CALIOP, DARDAR-MASK uses a decision tree to classify mixed-phase and differentiates it from highly concentrated ice particles, for a temperature range from $-40$ °C to $0$ °C (Ceccaldi et al., 2013). In some cases, highly concentrated ice particles areas are

**Table 4.** DARDAR-MASK v2.23 Classes. The phases currently processed in VarPy-mix are those indicated in bold.

| Number | Class |
|---|---|
| -2 | Presence of liquid unknown |
| -1 | Surface and subsurface |
| 0 | Clear sky |
| **1** | **Ice clouds** |
| **2** | **Spherical or 2D ice** |
| **3** | **Supercooled water** |
| **4** | **Supercooled water and ice** |
| 5 | Cold rain |
| 6 | Aerosol |
| 7 | Warm rain |
| 8 | Stratospheric clouds |
| **9** | **Highly concentrated ice particles** |
| **10** | **Top of convective towers** |
| 11 | Liquid clouds |
| 12 | Warm rain and liquid clouds |
| 13 | Cold rain and liquid clouds |
| 14 | Rain maybe mixed with liquid |
| **15** | **Multiple scattering due to supercooled water** |

**Table 5.** Cloud phases processed by VarPy-ice and VarPy-mix. Single group for VarPy-ice and two groups (ice and liquid) for VarPy-mix.

| Number | Class | VarPy-ice | VarPy-mix | |
|---|---|---|---|---|
| | | | Group "ice" | Group "liquid" |
| 1 | Ice clouds | ✓ | ✓ | |
| 2 | Spherical or 2D ice | ✓ | ✓ | |
| 3 | Supercooled water | | | ✓ |
| 4 | Supercooled water and ice | ✓ | ✓ | ✓ |
| 9 | Highly concentrated ice particles | ✓ | ✓ | |
| 10 | Top of convective towers | ✓ | ✓ | |
| 15 | Multiple scattering due to supercooled water | | | ✓ |

incorrectly classified as mixed-phase and need to be corrected for VarPy. These gates are then replaced by highly concentrated ice particles in the cloud phase classification.

## 2.4 Summary of the methodology

In the previous subsections, we described the principle and structure of the VarPy-mix method. Here, we summarize the five main key points of the method:

1. The radar reflectivity and the lidar backscatter measurements are the algorithm inputs. Their combination provide a cloud phase classification. This information is essential, as supercooled droplets and ice particles are not processed in the same way in our approach. We improve, correct and extend the classification to supercooled water (pure or in mixed-phase).

2. The state vector is composed of variables linked to both the measurements and the microphysical properties to be retrieved. We propose a state vector structure that allows us to simultaneously retrieves both ice particle and supercooled water droplet properties, either pure or in mixed-phase.

3. We assume that the lidar ratio of liquid water is constant with a value of 18.6 sr.

4. Based on the radar and lidar sensitivity, the ice part of the mixed-phase are retrieved with the radar signal and the liquid part with the lidar signal. Consequently, this influences the Jacobian structure, which is calculated by the radar and lidar forwards models.

5. The parameterization (errors, *a priori*, first guess, LUT, smoothing parameters, etc.) to retrieve ice microphysical properties comes from the VarPy-ice version. For supercooled water properties, a new parametrization is applied and a new LUT is created based on a log-normal distribution.

## 3 Example of retrieval and comparison with collocated *in situ* measurements

During the Arctic Study of Tropospheric Aerosol, Cloud and Radiation (ASTAR, Gayet et al., 2009; Ehrlich et al., 2009) campaign, four legs coming from the same flight were performed on the 7[th] of April 2007 over the ocean near Svalbard archipelago. The case presented in this study is one of the rare CloudSat-CALIPSO transect with collocated airborne *in situ* measurements of mixed-phase clouds. The *in situ* data from three probes is compared in this study to VarPy-mix retrievals. This comparison is possible because cloud detection as well as phase identification between DARDAR-MASK and *in situ* observations are in overall good agreement. Indeed, Mioche and Jourdan (2018) shows that 91 % of clear sky events and 86 % of the cloudy gates of DARDAR-MASK match with the Polar Nephelometer *in situ* probe from samples collected during the ASTAR 2007 and POLARCAT 2008 (see the Special Issue on POLARCAT in *Atmos. Chem. Phys.*) campaigns. The Polar Nephelometer can also be used to estimate the cloud phase observed (ice, liquid water and mixed-phase) thanks to thresholds on the asymmetry parameter $g$ (Jourdan et al., 2010). Using the Polar Nephelometer as a reference, Mioche and Jourdan (2018) shows that 61 % of DARDAR-MASK classification corresponding to ice phase match with Polar Nephelometer data, 67 % for liquid phase while 24 % for mixed-phase. This identification difference may be due to the temporal and spatial difference between satellite and *in situ* observations, or to the detection limit of supercooled water by lidar due to attenuation.

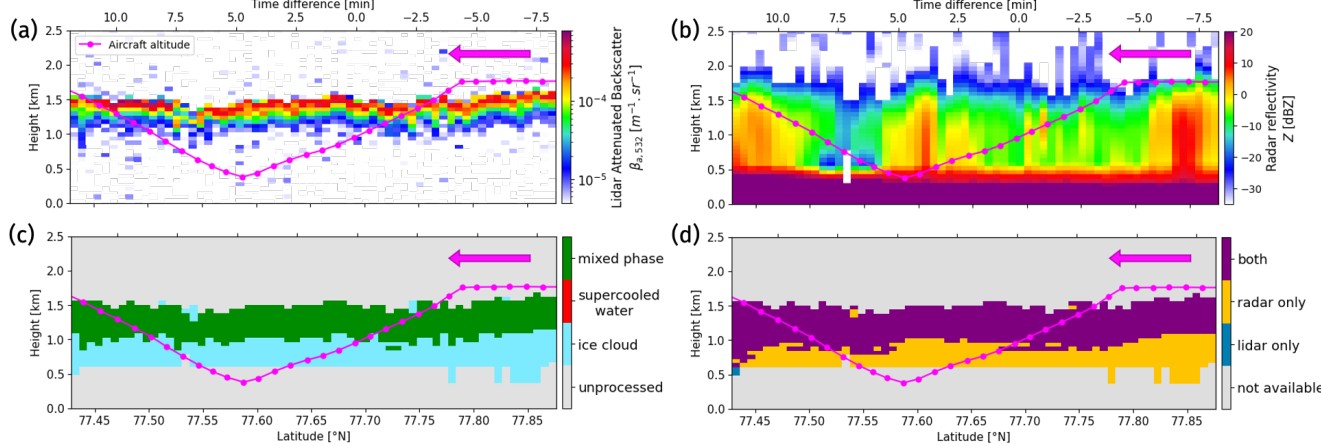

**Figure 2.** Selected profiles of CALIPSO attenuated backscatter (a), CloudSat reflectivity (b), processed cloud phase classification (c) and instrument synergy (d). The trajectory and direction of Polar-2 are shown by respectively magenta line and arrow.

### 3.1 Remote sensing and *in situ* measurements

For this comparison, the radar and lidar measurements and the classifications come from the DARDAR-MASK v2.23 product (Cazenave et al., 2019). The selected latitude range is shown in Fig. 2 panels, which present the profiles of the lidar backscatter
measurements (a), the radar reflectivity (b), the processed cloud phase classification (c) and the instrument flag to know which instrument is used for the retrieval (d). The strong lidar backscatter signal at the top of the cloud means that there is a large amount of small particles like supercooled water droplets. As the radar also detects particles in this part of the cloud this means that there are also ice particles. The processed cloud phase classification then shows the presence of an ice cloud with mixed-phase layer at the top. As presented in Fig. 2 (d), the mixed-phase is retrieved with both radar and lidar and the ice cloud
below is mainly retrieved with radar only, as the lidar is strongly attenuated and extinguished due to the supercooled water of the mixed-phase. As a result, note that the base of the supercooled liquid layer within the mixed-phased cloud cannot be determined unequivocally.

The three *in situ* instruments onboard the Polar-2 aircraft were the Cloud Particle Imager (CPI; Lawson et al., 1998), the Forward Scattering Spectrometer Probe (FSSP-100; Dye and Baumgardner, 1984; Gayet et al., 2007) and the Polar Neph-
elometer (PN; Gayet et al., 1997). As the aircraft was not flying exactly along the satellites trajectory, nor at the same time, the collocation is quite challenging. Among the four legs, the third one is temporally the closest to the satellites overpass with less than ten minutes delay (shown on the top x axis of Fig. 2). We focus this study on this leg to compare VarPy-mix retrievals to the *in situ* measurements. The altitude of the aircraft is shown by the magenta line in Fig. 2, where each point corresponds to a 30 second averaged probe measurements and the magenta arrow indicates the direction of the flight. As the aircraft flew above
the cloud before going inside the cloud and passing through the mixed-phase layer twice, we have then a vertical description of the cloud and the comparison with VarPy-mix retrieval is more complete.

The size range sensitivity of each probe is presented in Fig. 3. We assume here that the CPI provides information on ice particles, while the FSSP provides information on liquid water. We cannot exclude that the FSSP also detects secondary ice particles (Costa et al., 2017) or could be more likely contaminated by ice crystal shattered on the instrument tips. However, Costa et al. (2017) showed that secondary ice particles are not frequent in Arctic mixed-phase clouds. The temperature range at which cloud were probe (between $-21\,^{\circ}$C and $-14\,^{\circ}$C) does not point towards possible secondary ice production mechanisms (above $-10\,^{\circ}$C). Additionally, Febvre et al. (2012) showed that when ice crystals are measured by the FSSP, the asymmetry parameter measured by the PN decreases compared to what would be expected for water droplets only. In our case study, the asymmetry parameter $g$ is mostly greater than $0.84$ in the upper cloud layer which is indicative of a layer composed quasi-exclusively of water droplets. Consequently, we are quite confident that the presence of small ice crystals does not significantly impact the results.

For this study, we derive the ice cloud extinction $\alpha_{\mathrm{CPI}}$, the ice water content $\mathrm{IWC}_{\mathrm{CPI}}$, the ice effective radius $r_{e,\mathrm{CPI}}$ and the ice number concentration $N_{\mathrm{CPI}}$ from the CPI. The mass-size relationship to calculate $\mathrm{IWC}_{\mathrm{CPI}}$ is given by Equation 17 (model B for $0.2\,\mathrm{kg.m}^{-2}$ in Leinonen and Szyrmer, 2015). It corresponds to moderate riming and gives the best agreement over the whole flight.

$$m = 0.033 \times D^{1.94} \tag{17}$$

The liquid cloud extinction $\alpha_{\mathrm{FSSP}}$, liquid water content $\mathrm{LWC}_{\mathrm{FSSP}}$, the liquid effective radius $r_{e,\mathrm{FSSP}}$ and the liquid number concentration $N_{\mathrm{FSSP}}$ are provided by the FSSP. Both $r_{e,\mathrm{CPI}}$ and $r_{e,\mathrm{FSSP}}$ are calculated according to the following formula (Foot, 1988):

$$r_e = \frac{3}{2}\frac{\mathrm{WC}}{\rho\alpha} \tag{18}$$

where $r_e$ is the ice ($r_{e,\mathrm{CPI}}$) or liquid ($r_{e,\mathrm{FSSP}}$) effective radius, WC is the ice ($\mathrm{IWC}_{\mathrm{CPI}}$) or liquid ($\mathrm{LWC}_{\mathrm{FSSP}}$) water content, $\rho$ is the density of ice ($917\,\mathrm{kg.m}^{-3}$) or water ($1000\,\mathrm{kg.m}^{-3}$) and $\alpha$ the ice ($\alpha_{\mathrm{CPI}}$) or liquid ($\alpha_{\mathrm{FSSP}}$) extinction.

By summing extinctions, water contents and concentrations from both instruments, the total extinction $\alpha_{\mathrm{CPI+FSSP}}$, the total water content $\mathrm{TWC}_{\mathrm{CPI+FSSP}}$ and the total number concentration $N_{\mathrm{CPI+FSSP}}$ can be obtained. In addition, the PN provides the total extinction $\alpha_{\mathrm{PN}}$. These *in situ* measurements are shown in the next subsection in Fig. 4, 5, 6 and 7, and are detailed in the comparison in Sect. 3.3. The uncertainties of extinctions, water contents, number concentrations and asymmetry parameters are presented in Table 6 (Mioche et al., 2017).

### 3.2 VarPy-mix retrievals

The cloud phase classification has been adapted by eroding isolated supercooled gates. In this study, we chose to retrieve ice properties with the HC LUT. This implies that $A_{HC}$, $B_{HC}$ and $\gamma_{HC}$ values are used for the *a priori* and first guess values of $\ln(N_{0,\mathrm{ice}}^{*})$. For the liquid LUT, the only parameter of the size distribution that can vary is the geometric standard deviation $\sigma$. Fielding et al. (2014, 2015) set this value to $\sigma = 0.3 \pm 0.1$ and Frisch et al. (1995) at $\sigma = 0.35$. We chose here to set the geometric standard deviation to $\sigma = 0.3$.

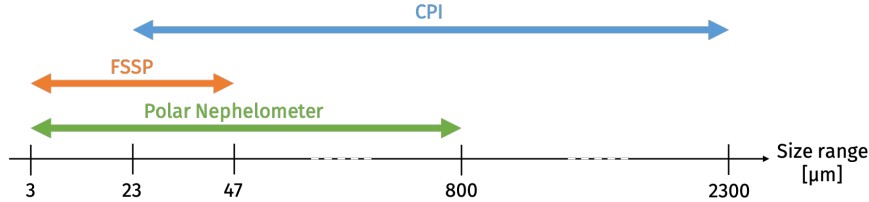

**Figure 3.** CPI, FSSP and PN range sensitivities.

**Table 6.** Uncertainties of cloud properties derived from CPI, FFSP and PN probes from Mioche et al. (2017).

| Properties | CPI | FSSP | PN |
|---|---|---|---|
| Extinction $\alpha$ | 55 % | 35 % | 25 % |
| Water Content (IWC or LWC) | 60 % | 20 % | - |
| Number concentration $N$ | 50 % | 10 % | - |
| Asymmetry parameter $g$ | - | - | 4 % |

First, the liquid and ice extinctions retrieved by VarPy-mix are shown by curtain in Fig. 4 (a) and (b) respectively, and are used to access more liquid and ice properties via LUTs. Figures 5 (a) and (b) show the LWC and IWC, Fig. 6 (a) and (b) show $r_{e,\mathrm{liq}}$ and $r_{e,\mathrm{ice}}$ and Fig. 7 (a) and (b) show $N_{\mathrm{liq}}$ and $N_{\mathrm{ice}}$. For each microphysical properties, the ice and liquid parts are retrieved, according to the classification. For the ice cloud between $0.5$ and $1$ km, only the ice properties are available. Ice and liquid properties are both retrieved for the mixed-phase gates.

Table 7 presents the mean values in all selected pixels of all retrieved properties. These values allow us to observe trends for each variable. The extinction of liquid droplets is stronger than ice particles by a factor of 7. The same trends is observed between LWC and IWC with average values $30$ % larger for LWC. The ice particles are larger than liquid droplets by a factor of 5 for the mean values. The liquid number concentration is much higher than ice number concentration by a factor $10^3$. All retrieved variables can be compared with *in situ* measurements. For extinction, water content and concentration, it is possible to sum the ice and liquid variables to obtain the total extinction $\alpha_{\mathrm{tot}}$ (curtain in Fig. 4 (c)), the total water content TWC (curtain in Fig. 5 (c)) and the total number concentration $N_{\mathrm{tot}}$ (curtain in Fig. 7 (c)).

### 3.3 Comparison

The retrieved total extinction of the mixed-phase layer is higher than the ice layer due to the presence of supercooled droplets. The extinctions from the CPI and FSSP have been summed in order to compare it to the total extinction of VarPy-mix and the one from the PN. These results are presented in Fig. 4 (c) by the dots and share the same colorscale as VarPy-mix curtain. Above the cloud, where it is clear sky for VarPy-mix (coming from radar and lidar measurements and classifications), the PN detects no particle and CPI+FSSP total extinction is very low ($10^{-8}$ m$^{-1}$ for the FSSP). Inside the cloud, we can observe the

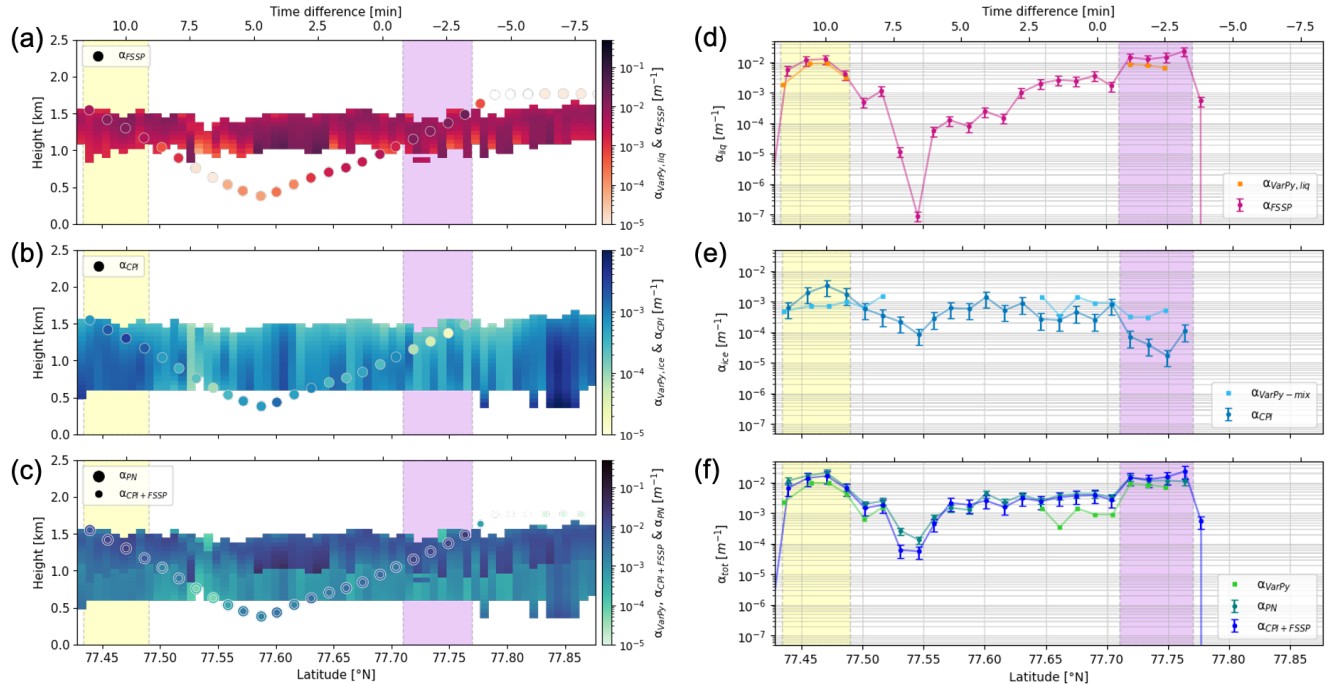

**Figure 4.** The panels (a) to (c) represent the liquid (a), ice (b) and total (c) extinctions from VarPy-mix retrievals (curtain) and *in situ* probes (dots) regarding the latitude and the height. The panels (d) to (f) represent the liquid (d), ice (e) and total (f) extinctions from VarPy-mix retrievals and *in situ* probes regarding the latitude. The error bars of *in situ* measurements (uncertainties from Table 6) are displayed in panels (d) to (f). The yellow and purple shading represents the latitude range where mixed-phase retrievals are compared with *in situ*.

same trend between VarPy-mix retrieval and probe results, which is mainly different between ice only area and the mixed-phase layer.

In order to provide a more detailed comparison, we keep only the gates from VarPy-mix that are closest to the *in situ* measurements. Figure 4 (f) displays by dots and lines the total extinction from the probes and from VarPy-mix. The points corresponding to the mixed-phase layer are highlighted on all figures by yellow and purple vertical shading and the others correspond to the ice cloud. Between $77.52$ and $77.64°$ N in Fig. 4 (f), there is no data for VarPy-mix because these points corresponds to ground clutter (ocean) area for the radar. The extinction for mixed-phase is higher than for the ice cloud, and this trend is observed for all results. In general, VarPy-mix total extinction is lower than total extinction from probes, especially in regions where cloud phase classification is defined as ice. In these regions the FSSP detects liquid droplets while CALIOP signal cannot be used because of the attenuation (extinguished). This can explain why $\alpha_{\text{VarPy}}$ is lower than $\alpha_{\text{CPI+FSSP}}$.

In mixed-phase layer, IWC and LWC are both retrieved by VarPy-mix and can be compared to *in situ* data, respectively from the CPI and the FSSP. The TWC is also used in this comparison. The results are shown in all panels of Fig. 5. In both regions of mixed-phase measurements, the LWC retrieved by VarPy-mix is between $2 \times 10^{-2}$ and $2 \times 10^{-1}$ $\text{g.m}^{-3}$ and agree well with the FSSP. Regarding the IWC, both CPI and VarPy-mix retrieve similar trends in these regions. In the region below, due to the

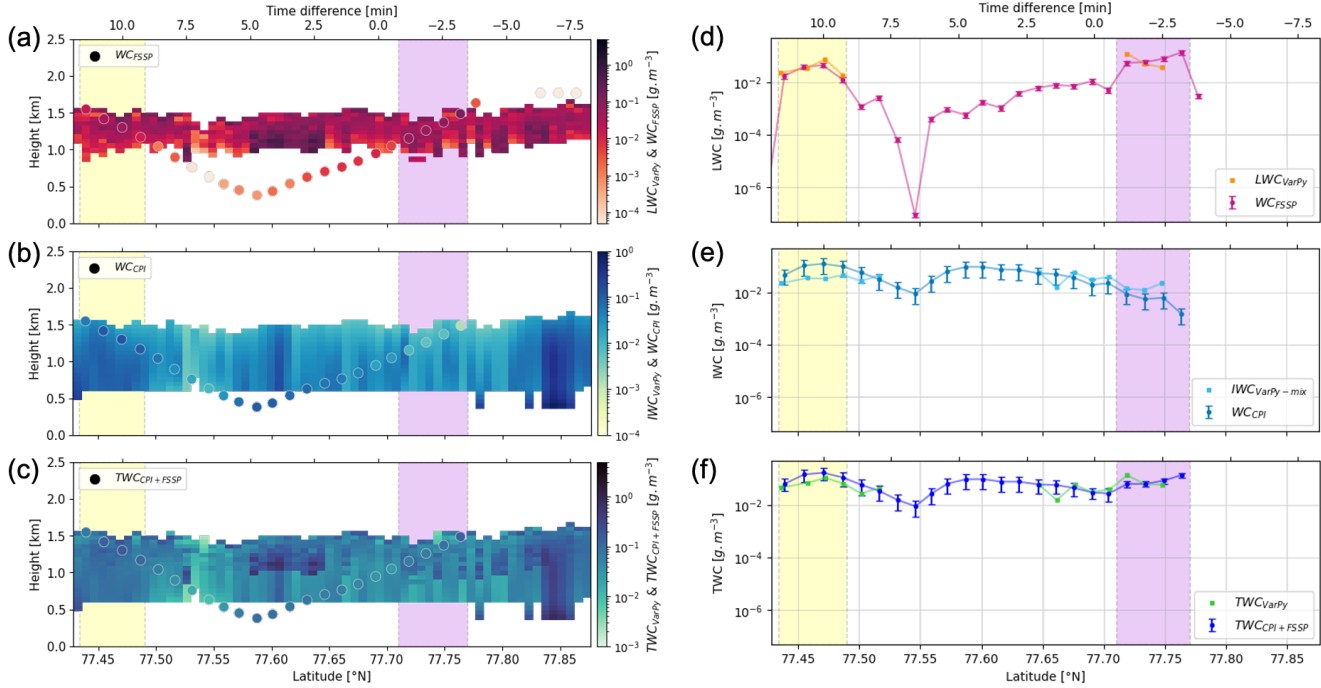

**Figure 5.** As Fig. 4 for LWC and IWC and TWC.

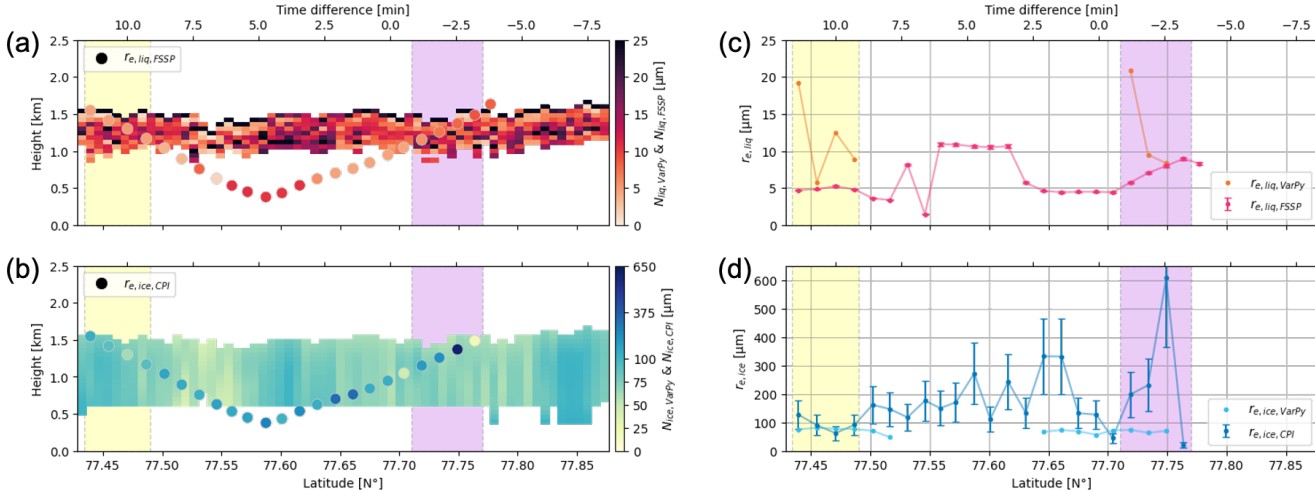

**Figure 6.** As Fig. 4 for $r_{e,\mathrm{liq}}$ and $r_{e,\mathrm{ice}}$.

extinction of the lidar signal, only ice properties are retrieved by VarPy but the FSSP detects also liquid in this region which impacts the comparison of the TWC. For that reason, we only compare in this region the IWC retrieved by VarPy-mix to the

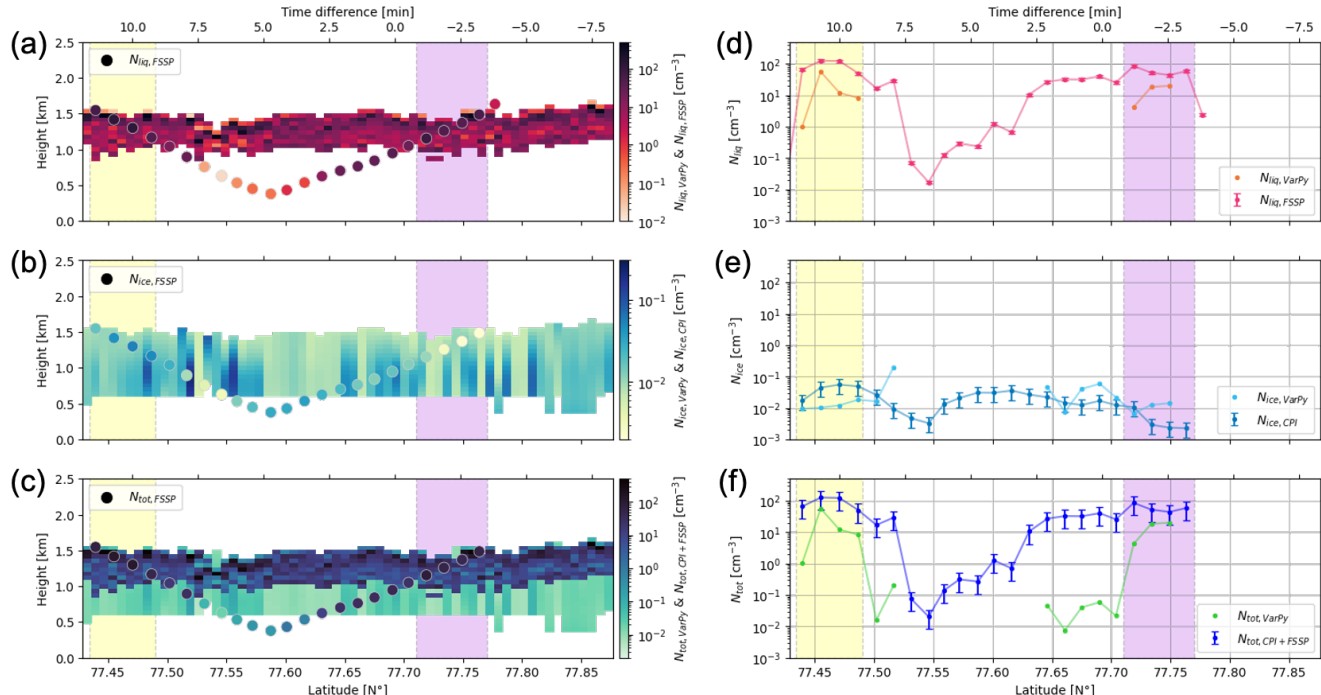

**Figure 7.** As Fig. 4 for $N_{\text{liq}}$, $N_{\text{ice}}$ and $N_{\text{tot}}$.

**Table 7.** Mean values of retrieved properties.

| Properties | Mean |
|:---:|:---:|
| $\alpha_{\text{ice}}$ | $1.03 \times 10^{-3}$ m$^{-1}$ |
| $\alpha_{\text{liq}}$ | $7.28 \times 10^{-3}$ m$^{-1}$ |
| $\alpha_{\text{tot}}$ | $4.91 \times 10^{-3}$ m$^{-1}$ |
| IWC | $5.32 \times 10^{-2}$ g.m$^{-3}$ |
| LWC | $6.89 \times 10^{-2}$ g.m$^{-3}$ |
| TWC | $8.99 \times 10^{-2}$ g.m$^{-3}$ |
| $r_{e,\text{ice}}$ | 75.2 µm |
| $r_{e,\text{liq}}$ | 13.5 µm |
| $N_{\text{ice}}$ | $2.01 \times 10^{-2}$ cm$^{-3}$ |
| $N_{\text{liq}}$ | $3.73 \times 10^{1}$ cm$^{-3}$ |
| $N_{\text{tot}}$ | $1.99 \times 10^{1}$ cm$^{-3}$ |

IWC from CPI, which are close to each other (40 % mean percent error). The region between 77.52 and 77.64° N cannot be compared, for the same reason as for the extinction.

**Table 8.** Mean absolute error and mean percent error regarding *in situ* for each property.

| Properties | Mean values for VarPy-mix selected gates | Mean values for *in situ* | Mean absolute error | Mean percent error |
|---|---|---|---|---|
| $\alpha_{\text{ice}}$ | $8.1 \times 10^{-4}$ m$^{-1}$ | $6.8 \times 10^{-4}$ m$^{-1}$ | $7.2 \times 10^{-4}$ m$^{-1}$ | 398 % |
| $\alpha_{\text{liq}}$ | $6.7 \times 10^{-3}$ m$^{-1}$ | $3.3 \times 10^{-3}$ m$^{-1}$ | $4.3 \times 10^{-3}$ m$^{-1}$ | 39 % |
| $\alpha_{\text{tot}}$ (CPI+FSSP) | $4.2 \times 10^{-3}$ m$^{-1}$ | $4.1 \times 10^{-3}$ m$^{-1}$ | $3.4 \times 10^{-3}$ m$^{-1}$ | 50 % |
| $\alpha_{\text{tot}}$ (PN) | $4.2 \times 10^{-3}$ m$^{-1}$ | $6.2 \times 10^{-3}$ m$^{-1}$ | $4.2 \times 10^{-3}$ m$^{-1}$ | 56 % |
| IWC | $2.9 \times 10^{-2}$ g.m$^{-3}$ | $3.4 \times 10^{-2}$ g.m$^{-3}$ | $5.0 \times 10^{-2}$ g.m$^{-3}$ | 75 % |
| LWC | $2.6 \times 10^{-2}$ g.m$^{-3}$ | $5.2 \times 10^{-2}$ g.m$^{-3}$ | $1.4 \times 10^{-2}$ g.m$^{-3}$ | 49 % |
| TWC | $3.0 \times 10^{-2}$ g.m$^{-3}$ | $6.0 \times 10^{-2}$ g.m$^{-3}$ | $4.7 \times 10^{-2}$ g.m$^{-3}$ | 39 % |
| $r_{e,\text{ice}}$ | 69.7 µm | 177.5 µm | 128.2 µm | 54 % |
| $r_{e,\text{liq}}$ | 12.2 µm | 5.56 µm | 6.40 µm | 122 % |
| $N_{\text{ice}}$ | $3.40 \times 10^{-2}$ cm$^{-3}$ | $2.02 \times 10^{-2}$ cm$^{-3}$ | $3.24 \times 10^{-2}$ cm$^{-3}$ | 280 % |
| $N_{\text{liq}}$ | $1.73 \times 10^{1}$ cm$^{-3}$ | $2.59 \times 10^{1}$ cm$^{-3}$ | $6.10 \times 10^{1}$ cm$^{-3}$ | 77 % |
| $N_{\text{tot}}$ | 8.69 cm$^{-3}$ | $3.59 \times 10^{1}$ cm$^{-3}$ | $4.51 \times 10^{1}$ cm$^{-3}$ | 89 % |

The same comparison between VarPy-mix retrievals and *in situ* measurements can be done for effective radii and concentrations, and is illustrated in all panels of Fig. 6 and 7 respectively. We can see on panel (a) and (c) of Fig. 6 that the liquid effective radius retrieved by VarPy-mix is higher than that from the FSSP. On the other hand, the ice effective radius from VarPy-mix is very close to the CPI effective radius in the mixed-phase layer indicated by the yellow shading (panels (b) and (e)). However, the values retrieved by VarPy-mix is much lower for the mixed-phase region indicated by the purple shading. In

this region, the CPI gives ice effective radius between 200 and 600 µm while VarPy-mix retrieves values around 70 µm. This difference may be due to the mass-size relationships applied, which differ between VarPy-mix and *in situ* data. Regarding the concentrations (Fig. 7), VarPy-mix retrieved less concentrated liquid particles than the FSSP and follow the same trend. For ice number concentration, the values are lower for VarPy-mix in the mixed-phase layer indicated by the yellow shading and higher in the one indicated by purple shading. On panel (f) of Fig. 7, the same trend as for the total extinction is obtain with

higher values in the mixed-phase layer and very low values below it. The explanations are the same as for the extinction.

For all variables, the mean absolute error (the mean of the absolute difference between each value of VarPy-mix and *in situ*) and the mean percent error regarding *in situ* (the mean of the absolute difference between each value of VarPy-mix and *in situ* divided by *in situ* value and expressed as a percentage) are calculated and are presented in Table 8. The liquid extinction retrieved by VarPy-mix differs from *in situ* by 39 %, which is similar to *in situ* uncertainties (35 %), and is the closest to the

*in situ* measurements. On the contrary, the mean percent error of ice extinction is 398 %. This can be explained by the large difference around 77.75° N, shown by the purple shading. The uncertainties of *in situ* probes (Table 6) also need to be taken into account.

The comparison between VarPy-mix retrieval and the *in situ* measurements is limited for many reasons. First the collocation in space is not perfect which can lead to biases and restrain this study to one case. The Polar 2 aircraft flew almost exactly under

465 CloudSat and CALIPSO trajectory during the third leg by crossing it around $77.6°$ N. If we do not consider the measurement points above the clouds, the maximum spatial shifts are $1.68$ km around $77.44°$ N and $1.34$ km around $77.78°$ N. The temporal shift is also the best for the third leg with less than ten minutes between the two platforms. Nevertheless, the sampling volumes of the probes are much smaller than those of the remote sensing instruments. Moreover, the vertical ($60$ m) and horizontal ($1.4$ km) resolutions of VarPy-mix products are larger than the probe sampling volume.

Another source of bias comes from the partial synergy of the VarPy-mix version in the mixed-phase. Indeed, the retrieval relies more strongly on the *a priori* values than when both instruments are used to retrieve ice clouds properties (Delanoë et al., 2013). In addition, the ice cloud is mainly retrieved with radar only and therefore with *a priori* values, which are temperature dependent. Furthermore, the main advantage of VarPy is the ability to retrieve full cloud profiles.

## 4   Summary and discussion

In this paper, we propose a method to retrieve microphysical properties of ice, supercooled and mixed-phase clouds simultaneously, called VarPy-mix. This variational method can use radar reflectivity at 35 or 95 GHz and lidar backscatter at $532$ nm from spaceborne or satellite platform to get vertical profiles of extinctions, ice and liquid water contents, effective radii and number concentrations. Radar and lidar have different sensitivities to hydrometeors, due to their wavelength and therefore this difference is used to retrieve the mixed-phase. On the one hand, the lidar is very sensitive to small and highly concentrated

particles such as liquid droplets. On the other hand, the radar is sensitive to the particle size meaning that the signal is stronger for ice particles than for liquid droplets. Consequently, the ice clouds are retrieved with both instruments while the mixed-phase retrieval is divided in two parts: the ice particles are retrieved with the radar signal and the supercooled water with the lidar signal. Therefore, the retrieval relies strongly on *a priori* and errors values.

VarPy-mix is based on the algorithm VarCloud (Delanoë and Hogan, 2008) that retrieves ice cloud properties with radar,

lidar and radiometric data. The variational method is the same, but the structure of the algorithm has been adapted to deal with supercooled water and mixed-phase clouds. The main modification comes from the state vector composition, which is divided into two parts, allowing ice and liquid to be processed separately as required. All matrices related to the state vector have been adapted to it. Moreover, a new look up table dedicated to liquid properties has been created. Based on a log-normal droplet size distribution, it is used to retrieve supercooled water clouds and the liquid part of the mixed-phase. For the ice clouds and the

ice part of mixed-phase clouds, two look up tables are implemented: one is using the T-matrix and the mass-size relationship from Heymsfield et al. (2010) and the second one is a combination of Brown and Francis (1995) and Mitchell (1996) mass-size relationships. It is important to know the phase of the cloud in order to process each gate appropriately. For this, an intermediate classification has been implemented. It distinguishes between ice clouds, supercooled water and the mixed-phase. Adaptations have been made on this classification to improve the retrieval (reduce biases) and be consistent with the measurements.

The retrieved properties can be divided into two parts, with ice properties on one side and those of the liquid on the other. The results are vertical profiles of:

- ice and liquid extinctions, $\alpha_{\text{ice}}$ and $\alpha_{\text{liq}}$ [m$^{-1}$], which can be used to estimate total extinction $\alpha_{\text{tot}}$ ;

- ice and liquid water contents, IWC and LWC [kg.m$^{-3}$], which can be used to estimate total water content TWC ;

- ice and liquid number concentrations, $N_{\text{ice}}$ and $N_{\text{liq}}$ [m$^{-3}$], which can be used to estimate total number concentration $N_{\text{tot}}$ ;

- and ice and liquid effective radius, $r_{e,\text{ice}}$ and $r_{e,\text{liq}}$ [m].

The case presented in this study is a mixed-phase layer on top of ice boundary layer cloud, at high latitudes. Therefore, ice and liquid properties are retrieved on top of cloud and then ice properties for the ice cloud below. By comparing with *in situ* measurements from the ASTAR campaign, we can see that the cloud microphysical properties retrieved with VarPy-mix follow similar trends as *in situ* measurements and that the retrieval produces correct results.

However, this comparison shows some limitations. First, the lower part of the cloud is missing which compromises part of the comparison. In fact, the lidar is attenuated by the liquid droplets of the mixed-phase layer and extinguished after it. The radar does not detect down to the ocean because of the clutter and therefore cannot see the cloud base. Then, the spatial and temporal shifts between the aircraft and the satellites need to be taken into account, which are respectively less than $1.7\,\text{km}$ and $10\,\text{min}$ for the chosen case. Moreover, the sampling volume is not the same between *in situ* probes and CloudSat-CALIPSO ($60\,\text{m}$ vertical resolution). This makes it difficult to compare precisely a VarPy-mix gate to an *in situ* measurement. Finally, the ice cloud retrieval is mainly done with radar signal only and each part of the mixed-phase is also retrieved with single instrument. The retrieval in this case relies strongly on *a priori* values and the look up table, which includes some bias in the comparison with *in situ*. A fully synergistic retrieval would be much more reliable, with both instruments retrieving each part of the mixed-phase. Another possible improvement would be to optimize the *a priori* and first guess values for liquid with *in situ* statistics. Moreover, it is important to note that the relationships used to retrieve properties differ between VarPy-mix and *in situ* (e.g. the mass-size relationship to retrieve water contents). This has a particular impact on the comparison of ice effective radii. Finally, this study focuses on only one case of mixed-phase at high latitude, above the ocean, which does not allow to know how the algorithm would retrieve globally the mixed-phase and the supercooled water.

In this study, VarPy-mix is used to retrieve clouds properties with CloudSat and CALIPSO data. Nevertheless, it can also be apply to observations from other platforms. The french RALI airborne platform with the RASTA radar ($95\,\text{GHz}$) and LNG lidar (multi-wavelength, 355, 532 and $1064\,\text{nm}$, and High Spectral Resolution - HSR - at $355\,\text{nm}$) offers more possibilities for comparison with *in situ* measurements. During the RALI-THINICE campaign that took place in August 2022 near the Svalbard archipelago, the ATR42 from SAFIRE flown over and inside several mixed-phase cases, with RALI and *in situ* probes. VarPy-mix will be applied on RALI data and some comparison with *in situ* can be done to evaluate, validate and improve VarPy-mix parameterization. The same can be applied on other campaigns like HALO-(AC)[3] (Wendisch et al., 2024), which took place in March and April 2022 in the Arctic near the Svalbard archipelago. During this campaign, the HALO platform, consisting

of the radar MIRA (35 GHz) and the lidar WALES (532, 1064 nm and HSR at 532 nm) flew over mixed-phase clouds. Some collocation with aircraft performing *in situ* measurements was conducted during this campaign. More information on both

campaigns can be found in their website (RALI-THINICE, HALO-AC3). In addition, VarPy-mix could use data from the EarthCARE satellite platform, which is planned for launch in 2024 and includes Cloud Profiling Radar (CPR) at 94 GHz and ATmospheric LIDar (ATLID) at 355 nm with HSR.

*Data availability.* DARDAR-MASK v2.23 products are publicly available on the AERIS/ICARE website (http://www.icare.univ-lille1.fr/, last access: 1 December 2023).

*Author contributions.* CA developed the methodology, the new parameterization and implemented the new version of the algorithm, with support from JD, SG and FE. *In situ* data were provided by OJ, FT and GM and satellite data were provided by the AERIS/ICARE Data and Services Center. CA, JD and SG worked on defining the paper structure and content. CA wrote the paper with contributions from JD, SG, FE, OJ and FT.

*Competing interests.* The authors declare that they have no conflict of interest.

*Acknowledgements.*

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
