# Peer review of "Lidar-radar synergistic method to retrieve ice, supercooled and mixed-phase clouds properties"

_Atmospheric Measurement Techniques, 2023_

## Referee Comment (RC3)

Review of

**Lidar-radar synergistic method to retrieve ice, supercooled and mixed-phase clouds properties**

by Aubry et al.

**General**

In this study, a new, advanced algorithm is developed to classify mixed-phase clouds into liquid, mixed and fully glaciated clouds from remote sensing measurements. Further, microphysical and optical properties of the different phases can be retrieved from the measurements. This is an important step towards large-scale, detailed analysis of mixed-phase clouds, which have been difficult to detect but play an crucial role in cloud feedback to the climate.
I cannot express the innovation and importance of this work any better than Referee 3 and 2 have already done, so I like to say here only that I fully agree with them.

I also find the manuscript very well structured, fluently written and easy to understand. I am not an expert in remote sensing retrieval algorithms, but I was able to follow the explanations of the method and the innovations in it - but without being able to judge it well. Regarding the figures, I have some suggestions to make them easier to understand (see below the specific comments on the Figures).

There is only one more important point about which I have a question (see point 11 of the specific comments): the presented case study shows a cloud of about 1 km thickness. The information from lidar and radar together is only available in the upper half of the cloud, for the lower part there is no information from lidar.
Can satellite-borne lidar instruments generally only penetrate approx. 500 m deep into mixed phase clouds or is this determined by the thickness of the cloud in the upper part? Or, could thicker liquid clouds still be detected in the lower part, i.e. is only the lidar signal too weak in the present case?

All other points are minor and are listed in the specific comments. Overall, I recommend the manuscript for publication in AMT after minor revisions.

**Specific comments**

**1)** Page 11, line 219: ‚*Whereas, the coefficient applied to the liquid water is different and set to 10, since the thickness of the detected liquid layer is smaller than ice layer*.'

Is it generally the case that the thickness of the liquid layer is smaller than that of the ice layer?

**2)** Page 12, line 248ff: '*For this study we use the following log-normal relationship defined by Frisch et al. (1995).*'

Why you use the oldest of the three available parameterizations?

**3)** Page 13, line 278: '*Indeed, the radar is not used to retrieved the supercooled water neither in pure liquid clouds nor in mixed-phase clouds, ...*'

Typo.

**4)** Page 14, line 306f: '*On the other hand, where there is no radar signal and a strong lidar backscatter, it is categorized as "supercooled water"* … '

What is meant with ' where there is no radar signal ' ? I think that means that the radar could in principle measure but there is no signal? But what if the conditions are such that the radar cannot measure but there would be a signal ? Is such a cloud misclassified ? Does this happen?

**5)** Page 16, line 343f: '*...note that the base of the supercooled liquid layer within the mixed-phased cloud cannot be determined unequivocally.*'

From Figure 2 c, it is also visible that from comparison with the in situ measurements the lowest part of the cloud is not detetcted with the radar – or is this an uncertainty caused by the unperfact match between in situ and satellite observation ?

**6)** Page 17, line 354f: '*Consequently, the CPI gives information about the ice particles and the FSSP about liquid droplets.*'

The particles in the FSSP can also be 'secondary ice particles', which cannot be distinguished with the FSSP (see e.g. Costa et al. (2017). This should be mentionened here.

**7)** Page 17, line 355f: '.. *we take the ... ice water content IWC$_{CPI}$ from the CPI, ...*'

What mass-dimension relationship have you used to calculate IWC$_{CPI}$ ?
I found it a few lines later  (line 361 - HC mass-size relationship), but would find it more appropriate here.  And,  can you explain why you used this one ?

**8)** Page 18, line 371f: 'T*able 7 presents the mean values in all selected pixels of all retrieved properties.*'

Why not include the in-situ mean values in the table, at least for the time periods where both in-situ and remote sensing measurements are available ?   I think that would be useful.

**9)** Page 18, line 372ff: '*The extinction of liquid droplets is stronger than ice crystals by a factor of 7. The same trends is observed between LWC and IWC with average values 30 % larger for LWC. The ice crystals are larger than liquid droplets by a factor of 5 for the mean values. The liquid number concentration is much higher than ice number concentration by a factor 10$^3$ .*'

Should one see that from the figures? This would only be possible if you use the same color code in all panels (which is difficult, but not impossible), or at least the same limits in the color code scale (see also the comment b) on Figures 4, 5, right panels).

**10)** Page 20, line 381ff:  see comment 5).

**11)** Page 20, line 392f:  '*In these regions the FSSP detects liquid droplets while CALIOP signal cannot be used because of the attenuation (extinguished). This can explain why α$_{VarPy}$ is lower than α$_{CPI+FSSP}$.*'

I think that this effect deserves to be discussed in a little more detail, because this sounds as if liquid droplets in lower cloud layers are generally not detected. This raises the question of the limitations of the method in relation to the vertical extent of the cloud (see also genaral comment)?

However, the FSSP signal in this area is much weaker than in the mixed phase clouds above, so there are far fewer drops present. So a question is whether it would be possible to detect liquid drops with a concentration as high as in the mixed phase layer in the lower part of the cloud with the lidar?

Furthermore,  in the article by Costa et al. (2017) – who classified mixed phase clouds based on airborne in situ measurements –  it is shown in their Figure 8 that small cloud particles (up to 50 um, detetcted with a CAS instrument, which is similar to an FSSP)  are still present even in completely  glaciated  clouds. This is also listed in their Table 6.  It is not clear where these cloud  particles come from, but the clouds are still classified as glaciated because the number of liquid droplets is so small (<

~0.1 cm$^{-3}$) that they cannot be considered a liquid cloud.. This could be discussed here to show that the new classification method is applicable.

By the way, it would be also interesting to see $N_{liq}$ and $N_{ice}$ from the in situ observations - then one could see whether the number of droplets is so small that they can hardly be called a cloud.

**12)** Would it be an idea to look for other in situ cases for comparison ? The data base of Costa et al. (2017) might provide the in situ observations. Maybe not for this paper, but for future work ?

**Figures:**

**a) Figures 4, 5, 6:** I recommend to change the order of the panels, liquid at the top and ice below, just like in the atmosphere – this is more intuitive and thus easier for the reader.

**b) Figures 4, 5, right panels:** I recommend using the same y-axis scales for all three panels, so that the differences between the panels (phases) are better visible.

**References:**

**Costa,** A., Meyer, J., Afchine, A., Luebke, A., Günther, G., Dorsey, J. R., Gallagher, M. W., Ehrlich, A., Wendisch, M., Baumgardner, D., Wex, H., and Krämer, M.: Classification of Arctic, midlatitude and tropical clouds in the mixed-phase temperature regime,
Atmos. Chem. Phys., 17, 12219–12238, https://doi.org/10.5194/acp-17-12219-2017, 2017.

---

## Author Response (AR1)

**Lidar-radar synergistic method to retrieve ice, supercooled and mixed-phase clouds properties**
**Answers to referees**

First, we would like to thank the Anonymous Referee 3 for his/her times and very useful comments that helped us to significantly improve the paper. Please find hereafter our answers and related corrections in blue color. In addition, we have attached a updated version of the paper below. Figures labeled with a letter refer to those included in the answers, and those labeled with a number to those included in the updated version of the paper.

**Answers to Anonymous Referee 3**

The paper demonstrates an expansion of the radar-lidar synergy retrieval product varpy, the algorithm underlying the successful DARDAR-CLOUD and related CloudSat-CALIPSO retrieval products, to include supercooled liquid and mixed-phase clouds. The principle is that in clouds identified as mixed-phase by the DARDAR-MASK synergistic radar-lidar target classification produce, the radar observations are used to constrain the retrieval of ice, and the lidar measurements are used to constrain the retrieval of liquid cloud. The retrieval is demonstrated using a single 10 minute case study of high-latitude mixed-phase boundary layer clouds observed by a coordinated aircraft underflight of the A-Train during the ASTER campaign.

This is an important issue in spaceborne detection and retrieval of clouds, especially relating to cloud-radiation interactions, and an expansion of the family of DARDAR products is welcome, as is any chance to evaluate spaceborne retrievals with field campaigns.

Overall the paper is well-written, the algorithm is thoroughly described, and the plots are produced to a good standard and are easy to understand. Some of the limitations of this study are discussed by the authors ; however, there could be a clearer distinction between the limitations of the retrieval and those of the evaluation using the field campaign data, and both could do with expanded discussion to consider their implications, such as for application to a DARDAR product.

I recommend this paper for major revisions, subject to addressing the following comments.

**Major comments :**
- What are the macrophysical implications of the in-situ profile demonstrating that neither the radar nor the lidar are sampling the entire cloud ? Is there an indication from the in-situ data whether the cloud is mixed-phase through its entire depth, or does it transition to ice cloud ? This is obviously a wide-ranging issue relating to all spaceborne radar-lidar observations, but it seems there's a bit of information here from the in-situ measurements that's worth commenting on.

  ▶ Since the lidar signal is attenuated, it cannot detect the entire cloud profile and the radar is limited in this case down to an altitude of 500 m due to ground clutter. As a result, radar and lidar alone cannot determine the phase of the entire cloud profile. However, we can see in Fig. 4 that *in situ* measurements detect both crystals (with CPI and PN) and liquid water droplets (with FSSP and PN) down to around 400 m. Below this altitude, the cloud phase cannot be determined. Nevertheless, we can have information on the cloud phase using the asymmetry parameter $g$ from the polar nephelometer (PN) measurements. JOURDAN et al. 2003, 2010 have

shown that $g$ values measured by the PN are usually less than 0.8 in ice clouds and around 0.84 - 0.85 in liquid-phase clouds. Besides, GAYET et al. 2002 estimates the uncertainties of $g$ at 4 %. Figure A shows the cloud phase classification from remote sensing (curtain, panel (a)) and the asymmetry parameter $g$ (dots on panel (a) and line on panel (b)) from *in situ*. The parameter $g$ indicates the predominant presence of mixed-phase or liquid water between 1 and 1.5 km altitude (corresponding to the mixed-phase layer indicated by the radar-lidar classification). Below 1 km altitude, the phase is predominantly ice, with some areas of mixed phase, presuming the lesser presence of water droplets up to 400 m altitude.

[Figure]

FIGURE A – Panel a) : Cloud phase classification (curtain) and asymmetry parameter $g$ from the Polar Nephelometer (dots) as a function of the altitude and latitude. Panel b) : Asymmetry parameter $g$ as a function of the latitude. The asymmetry parameter on both panel share the same colorbar.

- Further to this, what are the implications for DARDAR-CLOUD's existing ice retrievals in mixed-phase cloud (Table 5) ? I'm not asking for a change of scope of the present paper ; however, it would be interesting to compare varpy-mix and varpy-ice in the present case study to explore the effect of interpreting lidar extinction in these two very different ways.

▶ We have processed the same case with VarPy-ice and the results are presented in Figure B (dark blue dots). In addition, Table A shows the mean absolute error and mean percent error for VarPy-mix and for VarPy-ice. It is difficult to make conclusions from this comparison, since the lidar is not used in this case by VarPy-ice (the retrieval is performed using radar data only). On the other hand, the retrieval of ice properties by VarPy-mix is constrained by the simultaneous retrieval of supercooled water properties.

| Properties | VarPy-mix | | VarPy-ice | |
| --- | --- | --- | --- | --- |
| | Mean absolute error VarPy-mix | Mean percent error VarPy-mix | Mean absolute error VarPy-ice | Mean percent error VarPy-ice |
| $\alpha_{\text{ice}}$ | $7.2{\times}10^{-4}\,\text{m}^{-1}$ | 398 % | $5.6{\times}10^{-4}\,\text{m}^{-1}$ | 204 % |
| IWC | $2.9{\times}10^{-2}\,\text{g.\,m}^{-3}$ | 75 % | $3.4{\times}10^{-2}\,\text{g.\,m}^{-3}$ | 62 % |

TABLE A – Comparison between VarPy-mix and VarPy-ice for the ASTAR case

[Figure]

TABLE B – Comparison between VarPy-mix and VarPy-ice for the ASTAR case

- In the purple part of the case study, where varpy-mix ice extinction is much higher than that measured in-situ but the IWC is about right, does this imply a significant bias in the retrieved ice effective radius?

   ▶ We have calculated the ice and liquid effective radii for *in situ* using Equations A and B (where $\rho_i$ and $\rho_w$ are the ice and water density respectively). Figure 6 shows the effective radii calculated for *in situ* and the values retrieved by VarPy-mix. We can see that the ice effective radius calculated from the *in situ* (CPI) are much larger than those retrieved by VarPy-mix, especially in the purple part. On the other hand, the liquid effective radius retrieved by VarPy-mix is larger than the values calculated for *in situ* (FSSP). It is important to note that the ice effective radius retrieved by VarPy-mix with the HC LUT cannot be higher than 164 $\mu$m, as shown in Figure C. As a result, it could not reach the effective radii measured by the CPI in the purple zone. To improve our comparison, we include in the revised manuscript (Section 3) the comparison between effective radii retrieved by VarPy-mix and those calculated from in situ data (in the same way as for extinction and water contents). In addition, following a question from Anonymous Referee 1, we also include a comparison of the concentrations retrieved by VarPy-mix and those calculated from CPI and FSSP PSDs.

$$r_{e,\text{ice}} = \frac{2}{3} \frac{\text{IWC}_{\text{CPI}}}{\alpha_{\text{CPI}} \cdot \rho_i} \qquad (A) \qquad r_{e,\text{liq}} = \frac{2}{3} \frac{\text{LWC}_{\text{FSSP}}}{\alpha_{\text{FSSP}} \cdot \rho_w} \qquad (B)$$

[Figure]

FIGURE 6 – The panels (a) and (b) represent the liquid (a) and ice (b) effective radii from VarPy-mix retrievals (curtain) and *in situ* probes (dots) regarding the latitude and the height. The panels (c) and (d) share the same ordinate axis and represent the liquid (c) and ice (d) effective radii from VarPy-mix retrievals and *in situ* probes regarding the latitude. The error bars of *in situ* measurements (uncertainties from Table 6) are displayed in panels (c) and (d). The yellow and purple shading represents the latitude range where mixed-phase retrievals are compared with *in situ*.

[Figure]

FIGURE C – Comparison between ice effective radius from BF and HC LUT.

- The evaluation of the retrieval is summarized using the mean absolute and mean percentage errors, indicating impressive performance in most cases ; however, more information is available here that could be expanded upon, especially relating to the bias. In addition to the table, is it possible to use a plot that shows either the PDF of the retrieval compared to the in-situ data, or the distribution of errors, to retain more of this information. ► We have calculated the percent error regarding in situ measurements for each value when VarPy-mix retrievals are available (seven values for liquid properties and fourteen values for ice properties). The results are shown in Figure D. It is interesting to look at these distributions to better assess the comparison. We can see that the errors seem to vary according to the cloud area analyzed. Unfortunately, the number of values is limited to draw conclusion. If the liquid water content retrieval is biased low, as appears to be the case here, is this a systematic bias or could we expect a different result in other case studies ? Please comment on these uncertainties.► Indeed, it is possible that this bias may change. However, we have not yet compared VarPy-mix retrievals with other in situ measurements. It would be interesting to carry out a study comparing VarPy-mix retrievals with other in situ measurements, for example on a larger data set and over different world regions.

[Figure]

FIGURE D – PDF for each variable.

- There is some ambiguity about the mass-size relation that is used. It is clear from the method section and conclusion that both the Heymsfield and Brown and Francis relations are implemented as LUTs in VarPy, and the coefficients for both are given. In the algorithm description we are told "For both VarPy-ice and -mix, both LUT are used to retrieve the ice properties" (L156-7) but in the result section the results are shown for the Heymsfield mass-size relation only (L360-2). Does the selection of the Heymsfield relation indicate that these results are better than those using Brown and Francis, and can the authors (or the reader) conclude anything from this? Is there any value in showing both results? Are there any cases, e.g. in the purple section where a large error in the retrieved effective radius is implied, where the other mass-size relation may improve the performance? It is a strength of the VarPy algorithm to be able to answer these questions, so please expand on this or resolve the ambiguity.

  ▶ The wording of the sentence (L156-7) is indeed unclear, and we propose replacing it with : "For both VarPy-ice and -mix, both LUT can used to retrieve the ice properties and one must be selected beforehand." For this study, we arbitrarily chose the HC LUT. It is also the most recently implemented LUT in VarPy (proposed by Cazenave et al. 2019 for the new DARDAR-CLOUD v3 configuration). Furthermore, when comparing the retrievals made with the HC LUT and the BF LUT (shown in Figure E page 7), it can be seen that the difference between the two retrievals is relatively minor for these case.

**Minor comments :**

- Fig 2 (a) there is some contouring around noisy features in the lidar backscatter that make this panel difficult to interpret

  ▶ The colorbar limits have been modified to avoid this inconvenience (see Figure 2).

[Figure]

FIGURE 2 – CALIPSO attenuated backscatter.

- There are minor differences between the figures that surprise and confuse when comparing features between plots :
  - In Fig. 6 the colorbar labels face the other way to those on all other figures.
  - In Fig. 6 the format of the time and latitude labels, and their positions, are different from Fig. 5. This makes it difficult to line up the features visually.

    ▶ We have replaced Figure 6 with two new figures, showing the comparison between VarPy-mix and in situ retrievals for effective radii (Figure 6) and concentrations (see Figure 7 in the following version of the manuscript).

[Figure]

FIGURE E – Comparison between the HC and the BF LUT.

**Références**

GAYET, Jean-François, Frédérique AURIOL, Andreas MINIKIN, Johan STRÖM, Marco SEIFERT, Radovan KREJCI, Andreas PETZOLD, Guy FEBVRE et Ulrich SCHUMANN (2002). "Quantitative measurement of the microphysical and optical properties of cirrus clouds with four different in situ probes : Evidence of small ice crystals". en. In : *Geophysical Research Letters* 29.24. _eprint : https ://onlinelibrary.wiley.com/doi/pdf/10.1029/2001GL014342, p. XXX-XXX. ISSN : 1944-8007. DOI : 10.1029/2001GL014342. URL : https://onlinelibrary.wiley.com/doi/abs/10.1029/2001GL014342 (visité le 18/11/2023) (cf. p. 2).

JOURDAN, Olivier, Sergey OSHCHEPKOV, Valery SHCHERBAKOV, Jean-Francois GAYET et Harumi ISAKA (sept. 2003). "Assessment of cloud optical parameters in the solar region : Retrievals from airborne measurements of scattering phase functions". en. In : *Journal of Geophysical Research : Atmospheres* 108.D18, 2003JD003493. ISSN : 0148-0227. DOI : 10.1029/2003JD003493. URL : https://agupubs.onlinelibrary.wiley.com/doi/10.1029/2003JD003493 (visité le 20/11/2023) (cf. p. 1).

JOURDAN, Olivier, Guillaume MIOCHE, Timothy J. GARRETT, Alfons SCHWARZENBÖCK, Jérôme VIDOT, Yu XIE, Valery SHCHERBAKOV, Ping YANG et Jean-François GAYET (2010). "Coupling of the microphysical and optical properties of an Arctic nimbostratus cloud during the ASTAR 2004 experiment : Implications for light-scattering modeling". en. In : *Journal of Geophysical Research : Atmospheres* 115.D23. _eprint : https ://onlinelibrary.wiley.com/doi/pdf/10.1029/2010JD014016. ISSN : 2156-2202. DOI : 10.1029/2010JD014016. URL : https://onlinelibrary.wiley.com/doi/abs/10.1029/2010JD014016 (visité le 07/11/2023) (cf. p. 1).

CAZENAVE, Quitterie, Marie CECCALDI, Julien DELANOË, Jacques PELON, Silke GROSS et Andrew HEYMSFIELD (mai 2019). "Evolution of DARDAR-CLOUD ice cloud retrievals : new parameters and impacts on the retrieved microphysical properties". English. In : *Atmospheric Measurement Techniques* 12.5. Publisher : Copernicus GmbH, p. 2819-2835. ISSN : 1867-1381. DOI : 10.5194/amt-12-2819-2019. URL : https://amt.copernicus.org/articles/12/2819/2019/ (visité le 18/10/2022) (cf. p. 6).

**Lidar-radar synergistic method to retrieve ice, supercooled and mixed-phase clouds properties**
**Answers to referees**

First, we would like to thank the Anonymous Referee 3 for his/her times and very useful comments that helped us to significantly improve the paper. Please find hereafter our answers and related corrections in blue color. In addition, we have attached a updated version of the paper below.

**Answers to Anonymous Referee 2**

The presented work addresses a significant gap in current climate and weather forecasting models by proposing a novel method for retrieving microphysical properties of mixed-phase clouds. Recognizing the limited representation of these clouds due to the complexity of underlying processes, the study leverages airborne or satellite radar and lidar measurements within an extended variational method. The key innovation lies in the simultaneous retrieval of ice and supercooled water properties, achieved by exploiting the distinct sensitivities of lidar and radar signals. The method is rigorously developed, considering the sensitivity of these instruments and relying on a robust a priori framework. Validation using DARDAR-MASK products and airborne in situ measurements demonstrates promising results, showcasing the method's potential to enhance our understanding of mixed-phase cloud processes. The work is well-structured, providing clear insights into the motivation, methodology, and validation approach, and it contributes significantly to the broader goal of improving climate and weather prediction models. Overall, the research exhibits a commendable quality, combining scientific rigor with practical applicability.

The abstract effectively introduces the paper's focus on addressing the challenges in representing mixed-phase clouds in climate and weather forecasting models. The proposed method for retrieving microphysical properties of mixed-phase clouds is clearly outlined. The use of airborne or satellite radar and lidar measurements, along with a variational method, enhances the understanding of cloud processes. The abstract sets a comprehensive context for the reader, emphasizing the importance of accurate representation of mixed-phase clouds in models. Section 2 is well-written and provides a comprehensive explanation of the methodology. In section 3 the introduction of the ASTAR campaign and the specific context of the study is clear and sets the stage effectively. The integration of in situ measurements and remote sensing data, especially the use of the Polar Nephelometer as a reference, enhances the robustness of the comparison. Section 4 appropriately acknowledges the limitations and challenges of the VarPy-mix methodology, such as missing data in the lower part of the cloud, spatial and temporal shifts, and biases in the retrieval process. This transparency enhances the scientific rigor of the study.

**Specific Comments :**
- The abstract is well-structured, progressing logically from the motivation to the proposed method and concluding with the validation approach. The use of technical terms is appropriate, but consider providing brief explanations for readers less familiar with terms like "attenuated lidar backscatter $\beta$" to enhance accessibility.
    ▶ We added in the introduction (L50-51) : "On one hand, the lidar measures the attenuated backscatter $\beta$ $[m^{-1}.sr^{-1}]$, which corresponds to the energy backscattered by the targets and is affected by the atmospheric transmission."

- Section 2 : In some sentences, the detailed technical content and the complex structure might pose a challenge for readers unfamiliar with the specific terminology or mathematical formulations. Consider breaking down a few of the more complex sentences for better readability without sacrificing technical accuracy.

   ▶ Several sentences have been split and these changes are indicated in blue in the following version of the manuscript (add at the end of this document). The lines concerned are listed hereafter : L55, L58, L72, L93-97 (add Eq. 1 and split in three sentences), L101, L230, L239, L258-265 (itemize)

- The scientific rigor of the proposed method is established in the abstract by clearly outlining the underlying assumptions and the methodology. It effectively communicates the dual sensitivity of lidar and radar in retrieving supercooled water and ice crystals, respectively.

   ▶ Thank you.

- Towards the end of Section 2, you might consider adding a brief summary or concluding paragraph that synthesizes the key aspects of the methodology. The connection between observations (Y) and the state vector (X) is effectively established.

   ▶ We have added Section 2.4 (L228-243).

- The validation approach using DARDAR-MASK products and airborne in situ measurements is a strong point, showcasing the method's applicability and performance. Mentioning the mean percent error provides a quantitative measure of the retrieval performance, enhancing the credibility of the results.
- The adaptation of the cloud phase classification for mixed-phase retrieval is well-motivated and aligns with the goal of correctly identifying each phase of the cloud.
- In general, section 3 contains a significant amount of technical detail, which is appropriate for a scientific paper, but ensure that the language remains accessible to the target audience.

   ▶ Thank you.

- Section 4 effectively summarizes the VarPy-mix methodology, discusses its strengths and limitations, and lays the groundwork for further discussion and conclusions. I would suggest to provide more context on the RALI-THINICE campaign and the HALO-(AC)3 campaign, as these are mentioned but not explained in detail.

   ▶ We have added information in the text (in blue, § L515-527) and refer to the campaign websites for more information (added in References). An Overview Paper for each measurement campaign is currently being submitted to the Bulletin of the American Meteorological Society for RALI-THINICE and Atmospheric Chemistry and Physics for HALO-(AC)³. As these papers are not yet accessible, we cannot refer to them here.

- Lines 23-25 : To gain a clearer understanding, could you provide more specific information on the largest differences observed in both time and space between remote sensing and in situ measurements ? Is there a quantitative measure or range that characterizes these differences or any strategies employed to address or mitigate these differences in the analysis ?

   ▶ We have added the following information in the abstract : "It is also important to note that temporal and spatial collocations are not perfect, with a maximum spatial shift of 1.68 km and a maximum temporal shift about ten minutes between the two platforms. In addition, the sensibility of remote sensing and in situ are not the same and in situ measurements uncertainties are between 25 % and 60 %."

- Lines 72-73 : I would recommend to introduce the VarPy-mix method earlier in the paper, possibly in the abstract or introduction, to provide readers with a clear understanding of the methodology from the outset. This can enhance the overall coherence and comprehension of the research, ensuring that the significance of VarPy-mix is highlighted prominently.

   ▶ "VarPy-mix" is now mentioned in the abstract.

**Technical Corrections :**

Consistency in Terminology :
- Ensure consistency in the use of terms such as "ice crystals" and "ice particles" for clarity.
  - ▶ We have replaced the terms "ice crystals" with "ice particles".

Abbreviations :
- Consider introducing abbreviations like IWC, LWC, and Ntot upon their first use in the abstract.
  - ▶ They are now all present in the abstract.
- Line 78 : "Additionally, this flexible algorithm can be apply on several radar-lidar platforms,"

**Lidar-radar synergistic method to retrieve ice, supercooled and mixed-phase clouds properties**
**Answers to referees**

First, we would like to thank the Anonymous Referee 1 for his/her times and very useful comments that helped us to significantly improve the paper. Please find hereafter our answers and related corrections in blue color. In addition, we have attached a updated version of the paper below. Figures labeled with a letter refer to those included in the answers, and those labeled with a number to those included in the updated version of the paper.

**Answers to Anonymous Referee 1**

In this study, a new, advanced algorithm is developed to classify mixed-phase clouds into liquid, mixed and fully glaciated clouds from remote sensing measurements. Further, microphysical and optical properties of the different phases can be retrieved from the measurements. This is an important step towards large-scale, detailed analysis of mixed-phase clouds, which have been difficult to detect but play an crucial role in cloud feedback to the climate.

I cannot express the innovation and importance of this work any better than Referee 3 and 2 have already done, so I like to say here only that I fully agree with them.

I also find the manuscript very well structured, fluently written and easy to understand. I am not an expert in remote sensing retrieval algorithms, but I was able to follow the explanations of the method and the innovations in it - but without being able to judge it well. Regarding the figures, I have some suggestions to make them easier to understand (see below the specific comments on the Figures).

There is only one more important point about which I have a question (see point 11 of the specific comments) : the presented case study shows a cloud of about 1 km thickness. The information from lidar and radar together is only available in the upper half of the cloud, for the lower part there is no information from lidar.

- Can satellite-borne lidar instruments generally only penetrate approx. 500 m deep into mixed phase clouds or is this determined by the thickness of the cloud in the upper part ? Or, could thicker liquid clouds still be detected in the lower part, i.e. is only the lidar signal too weak in the present case ?

▶ Satellite-borne lidar generally only penetrates up to approx. 500 m, depending on the supercooled water optical thickness, even if the supercooled water droplets are present on a 1 km layer. Since the satellite-borne lidar is located far from the clouds, its signal is partly attenuated before it reaches the cloud particles. In addition, the signal is strongly attenuated by supercooled droplets and satellite-borne lidar are too "weak" to penetrate deeper in the clouds (i.e. it is not the case for airborne lidar, which can still detect cloud particles after a mixed-phase layer, and sometimes airborne lidar can detect multiple mixed-phase layers, which is rarely the case for CALIPSO). Generally, lidar cannot penetrate optical thicknesses greater than 3. Figure A shows the total, ice and liquid optical thicknesses calculated from the total, ice and liquid extinctions retrieved by VarPy-mix, respectively. We can see that most of the total optical thickness comes from the liquid part. Furthermore, the thickness quickly rises above 3, making it difficult for lidar to penetrate deeper into the cloud.

All other points are minor and are listed in the specific comments. Overall, I recommend the manuscript for publication in AMT after minor revisions.

[Figure]

FIGURE A – Optical thickness retrieved by VarPy-mix (from total extinction).

**Specific comments**

1. Page 11, line 219 : ‚Whereas, the coefficient applied to the liquid water is different and set to 10, since the thickness of the detected liquid layer is smaller than ice layer.‘
   - Is it generally the case that the thickness of the liquid layer is smaller than that of the ice layer ?
   ▶ Since ice clouds are generally much more extended in height than supercooled water or mixed-phase layers, this coefficient is chosen according to this trend.

2. Page 12, line 248ff : ‘For this study we use the following log-normal relationship define**d** by Frisch et al. (1995).’
   - Why you use the oldest of the three available parameterizations ?
   ▶ The idea was to start with the log-normal distribution, which is more computationally convenient. Frisch et al. 1995 proposed this distribution and Fielding et al. 2015 adopted the same distribution, defining $r_0$ as the median radius instead of the modal radius. Since Frisch et al. 1995 proposed it and exposed the moments of the distribution, we cite this paper to explain the distribution we have chosen. Nevertheless, the parameterization of Fielding et al. 2014, 2015 are taking into account for the standard deviation of the distribution (lines 387-389 in the revised version). Besides, one possible perspective is to create a liquid look-up table from a gamma distribution and compare retrievals from the different liquid LUTs.

3. Page 13, line 278 : ‘Indeed, the radar is not used to retrieve**d** the supercooled water neither in pure liquid clouds nor in mixed-phase clouds, ...’
   - Typo
   ▶ Thank you, it has been corrected.

4. Page 14, line 306f : ‘On the other hand, where there is no radar signal and a strong lidar backscatter, it is categorized as "supercooled water" ... ‘
   - What is meant with ‘where there is no radar signal‘ ? I think that means that the radar could in principle measure but there is no signal ? But what if the conditions are such that the radar cannot measure but there would be a signal ? Is such a cloud misclassified ? Does this happen ?
   ▶ We mean that the particle are too small for the radar sensitivity, which consequently gives no radar signal. CloudSat (95 GHz) sensitivity does not allow detection of supercooled water. Regarding possible signal "mistakes", this can be corrected easily during the first steps of the radar data processing (we guess that in such cases, the radar signal will be very different than for real detection) and the misclassifications can be avoided. In the updated version, we have added (L314-315 : "On the other hand, where the radar does not detect particles (no radar signal) and the lidar backscatter is strong, ..."

5. Page 16, line 343f : '...note that the base of the supercooled liquid layer within the mixed-phased cloud cannot be determined unequivocally.'
   - From Figure 2 c, it is also visible that from comparison with the in situ measurements the lowest part of the cloud is not detetcted with the radar – or is this an uncertainty caused by the unperfact match between in situ and satellite observation ?
     ▶ The lower part is not detected by the radar because of the ground clutter (unwanted echoes from the ocean in that case). Even if the match was perfect, CloudSat has difficulty detecting below 1 km altitude due to the clutter phenomenon.

6. Page 17, line 354f : 'Consequently, the CPI gives information about the ice particles and the FSSP about liquid droplets.'
   - The particles in the FSSP can also be 'secondary ice particles', which cannot be distinguished with the FSSP (see e.g. Costa et al. (2017). This should be mentionened here.
     ▶ We provide more details by modifying the text with : "We assume here that the CPI provides information on ice particles, while the FSSP provides information on liquid water. We cannot exclude that the FSSP also detects secondary ice particles (COSTA et al. 2017) or could be more likely contaminated by ice crystal shattered on the instrument tips. However, COSTA et al. 2017 showed that secondary ice particles are not frequent in Arctic mixed-phase clouds. The temperature range at which cloud were probe (between −21 °C and −14 °C) does not point towards possible secondary ice production mechanisms (above −10 °C). Additionally, FEBVRE et al. 2012 showed that when ice crystals are measured by the FSSP, the asymmetry parameter measured by the PN decreases compared to what would be expected for water droplets only. In our case study, the asymmetry parameter $g$ is mostly greater than 0.84 in the upper cloud layer which is indicative of a layer composed quasi-exclusively of water droplets. Consequently, we are quite confident that the presence of small ice crystals does not significantly impact the results."
     In addition, we provide in Figure B the Particle Size Distribution of the FSSP. For each distribution, we have indicated the altitude and the asymmetry parameter $g$. The largest (and most numerous) drops are found in the upper layers (the two highest lines in each figure), with the corresponding $g$ (above 0.84). When $g$ falls below 0.84 (mixed-phase), drops are much less numerous.

[Figure]

FIGURE B – Particle Size Distribution (PSD) of the FSSP. The corresponding altitude and $g$ value are indicated in the legend.

7. Page 17, line 355f : '.. we take the ... ice water content IWC$_{\text{CPI}}$ from the CPI, ...'
   - What mass-dimension relationship have you used to calculate IWC$_{\text{CPI}}$? I found it a few lines later (line 361 - HC mass-size relationship), but would find it more appropriate here. And, can you explain why you used this one?
   ▶ The HC relationship corresponds to the chosen LUT for VarPy-mix. Line 361 has been corrected with : "In this study, we chose to retrieve ice properties with the HC LUT." (it is lines 385-386 in the revised manuscript). Nevertheless, the mass-size relationship to calculate the ice particle properties is given by Equation A (model B for 0.2 kg.m$^{-2}$ in LEINONEN et SZYRMER 2015). It corresponds to moderate riming and gives the best agreement over the whole flight. We have included this information in the new manuscript version (L 390-393).

$$m = 0.033D^{1.94} \tag{A}$$

8. Page 18, line 371f : 'Table 7 presents the mean values in all selected pixels of all retrieved properties.'
   - Why not include the in-situ mean values in the table, at least for the time periods where both in-situ and remote sensing measurements are available? I think that would be useful.
   ▶ The idea behind Table 7 was to present only the trends obtained for VarPy-mix retrievals, independently of the comparison with *in situ*. Instead, we propose to extend Table 8 with the mean *in situ* values and the mean VarPy-mix values for the selected gates that are spatially closest to the *in situ* data (corresponding to the values of the right-hand panels in Figures 4 and 5). We have also added to this table the values for concentrations (see answer to question 11) and effective radii (following a question from Anonymous Referee 3, we have calculated the in situ effective radii and added the comparison to the updated version of the manuscript).

TABLE 8 – Mean absolute error and mean percent error regarding *in situ* for each property.

| Properties | Mean values for VarPy-mix selected gates | Mean values for *in situ* | Mean absolute error | Mean percent error |
|---|---|---|---|---|
| $\alpha_{\text{ice}}$ | $8.1 \times 10^{-4}$ m$^{-1}$ | $6.8 \times 10^{-4}$ m$^{-1}$ | $7.2 \times 10^{-4}$ m$^{-1}$ | 398 % |
| $\alpha_{\text{liq}}$ | $6.7 \times 10^{-3}$ m$^{-1}$ | $3.3 \times 10^{-3}$ m$^{-1}$ | $4.3 \times 10^{-3}$ m$^{-1}$ | 39 % |
| $\alpha_{\text{tot}}$ (CPI+FSSP) | $4.2 \times 10^{-3}$ m$^{-1}$ | $4.1 \times 10^{-3}$ m$^{-1}$ | $3.4 \times 10^{-3}$ m$^{-1}$ | 50 % |
| $\alpha_{\text{tot}}$ (PN) | $4.2 \times 10^{-3}$ m$^{-1}$ | $6.2 \times 10^{-3}$ m$^{-1}$ | $4.2 \times 10^{-3}$ m$^{-1}$ | 56 % |
| IWC | $2.9 \times 10^{-2}$ g m$^{-3}$ | $3.4 \times 10^{-2}$ g m$^{-3}$ | $5.0 \times 10^{-2}$ g m$^{-3}$ | 75 % |
| LWC | $2.6 \times 10^{-2}$ g m$^{-3}$ | $5.2 \times 10^{-2}$ g m$^{-3}$ | $1.4 \times 10^{-2}$ g m$^{-3}$ | 49 % |
| TWC | $3.0 \times 10^{-2}$ g m$^{-3}$ | $6.0 \times 10^{-2}$ g m$^{-3}$ | $4.7 \times 10^{-2}$ g m$^{-3}$ | 39 % |
| $r_{e,\text{ice}}$ | 69.7 $\mu$m | 177.5 $\mu$m | 128.2 $\mu$m | 54 % |
| $r_{e,\text{liq}}$ | 12.2 $\mu$m | 5.56 $\mu$m | 6.40 $\mu$m | 122 % |
| $N_{\text{ice}}$ | $3.40 \times 10^{-2}$ cm$^{-3}$ | $2.02 \times 10^{-2}$ cm$^{-3}$ | $3.24 \times 10^{-2}$ cm$^{-3}$ | 280 % |
| $N_{\text{liq}}$ | $1.73 \times 10^{1}$ cm$^{-3}$ | $2.59 \times 10^{1}$ cm$^{-3}$ | $6.10 \times 10^{1}$ cm$^{-3}$ | 77 % |
| $N_{\text{tot}}$ | 8.69 cm$^{-3}$ | $3.59 \times 10^{1}$ cm$^{-3}$ | $4.51 \times 10^{1}$ cm$^{-3}$ | 89 % |

9. Page 18, line 372ff : 'The extinction of liquid droplets is stronger than ice crystals by a factor of 7. The same trends is observed between LWC and IWC with average values 30 % larger for LWC. The ice crystals are larger than liquid droplets by a factor of 5 for the mean values. The liquid number concentration is much higher than ice number concentration by a factor $10^3$ .'
   - Should one see that from the figures? This would only be possible if you use the same color code in all panels (which is difficult, but not impossible), or at least the same limits in the color code scale (see also the comment b) on Figures 4, 5, right panels).
   ▶ We propose the following rewording : "Table 7 presents the mean values in all selected pixels of all retrieved properties. These values allow us to observe trends for each variable. The extinction of liquid droplets is stronger than...". In addition, we use the same y axis scale for the three right panels of Figure 4 and 5. (Using the same scale color scale for the left panel curtains is more difficult.)

10. Page 20, line 381ff :
    - see comment 5).

11. Page 20, line 392f : 'In these regions the FSSP detects liquid droplets while CALIOP signal cannot be used because of the attenuation (extinguished). This can explain why $\alpha_{\text{VarPy}}$ is lower than $\alpha_{\text{CPI+FSSP}}$.'
    - I think that this effect deserves to be discussed in a little more detail, because this sounds as if liquid droplets in lower cloud layers are generally not detected. This raises the question of the limitations of the method in relation to the vertical extent of the cloud (see also genaral comment) ? ▶ This is indeed a problem that we mention in the conclusion on lines 503-504 (updated manuscript) :"First, the lower part of the cloud is missing which compromises part of the comparison. In fact, the lidar is attenuated by the liquid droplets of the mixed-phase layer and extinguished after it. The radar does not detect down to the ocean because of the clutter and therefore cannot see the cloud base." However, the FSSP signal in this area is much weaker than in the mixed phase clouds above, so there are far fewer drops present. So a question is whether it would be possible to detect liquid drops with a concentration as high as in the mixed phase layer in the lower part of the cloud with the lidar ? ▶ This is very rare with CALIPSO, but possible with airborne lidars. Furthermore, in the article by Costa et al. (2017) – who classified mixed phase clouds based on airborne *in situ* measurements – it is shown in their Figure 8 that small cloud particles (up to 50 um, detetcted with a CAS instrument, which is similar to an FSSP) are still present even in completely glaciated clouds. This is also listed in their Table 6. It is not clear where these cloud particles come from, but the clouds are still classified as glaciated because the number of liquid droplets is so small ($< \sim 0.1$ cm-3) that they cannot be considered a liquid cloud. This could be discussed here to show that the new classification method is applicable. By the way, it would be also interesting to see Nliq and Nice from the *in situ* observations - then one could see whether the number of droplets is so small that they can hardly be called a cloud.

      ▶ We calculated liquid, ice and total number concentrations from the PSDs of each probe (CPI and FSSP). The results are shown in Figure 7. Panels (a) and (d) show that the FSSP detects highly concentrated particles ($\sim 10^2$ cm$^{-3}$) in the yellow and purple zones. The values remain quite high in some areas below the mixed-phase layer delimited by remote sensing. Values fall below 10 cm$^{-3}$ between 77.52 and 77.62°N only. The question of layer thickness detection is therefore relevant and should be taken into account in the results of our method.

      It is also interesting to look at the asymmetry parameter $g$. JOURDAN et al. 2003, 2010 have shown that $g$ values measured by the PN are usually less than 0.8 in ice clouds and around 0.84 - 0.85 in liquid-phase clouds. Besides, GAYET et al. 2002 estimates the uncertainties of $g$ at 4 %. Figure C shows the cloud phase classification from remote sensing (curtain, panel (a)) and the asymmetry parameter $g$ (dots on panel (a) and line on panel (b)) from *in situ*. The parameter $g$ indicates the predominant presence of mixed-phase or liquid water between 1 and 1.5 km altitude (corresponding to the mixed-phase layer indicated by the radar-lidar classification). Below 1 km altitude, the phase is predominantly ice, with some areas of mixed phase, presuming the lesser presence of water droplets up to 400 m altitude.

[Figure]

FIGURE 7 – The panels (a) to (c) represent the liquid (a), ice (b) and total (c) number concentrations from VarPy-mix retrievals (curtain) and *in situ* probes (dots) regarding the latitude and the height. The panels (d) to (f) share the same ordinate axis and represent the ice (d), liquid (e) and total (f) number concentration from VarPy-mix retrievals and *in situ* probes regarding the latitude. The error bars of *in situ* measurements (uncertainties from Table 6) are displayed in panels (d) to (f). The yellow and purple shading represents the latitude range where mixed-phase retrievals are compared with *in situ*.

[Figure]

FIGURE C – Panel a) : Cloud phase classification (curtain) and asymmetry parameter *g* from the Polar Nephelometer (dots) as a function of the altitude and latitude. Panel b) : Asymmetry parameter *g* as a function of the latitude. The asymmetry parameter on both panel share the same colorbar.

12. Would it be an idea to look for other in situ cases for comparison? The data base of Costa et al. (2017) might provide the in situ observations. Maybe not for this paper, but for future work?

▶ It is a good idea to compare VarPy-mix retrievals with other *in situ* measurements. The different scenarios (according to latitude, season, etc.) analyzed with *in situ* constitute a database that could be compared with similar scenarios processed with VarPy-mix. For example, this would enable us to determine the behavior of VarPy-mix at different latitudes, and to assess these retrievals with in situ data. The best would be to have collocated data between CloudSat/CALIPSO and data from Costa et al. 2017.

**Figures :** a) Figures 4, 5, 6 : I recommend to change the order of the panels, liquid at the top and ice below, just like in the atmosphere – this is more intuitive and thus easier for the reader.

b) Figures 4, 5, right panels : I recommend using the same y-axis scales for all three panels, so that the differences between the panels (phases) are better visible.

▶ The Figures 4 and 5 have been corrected.

[Figure]

FIGURE 4 – The panels (a) to (c) represent the ice (a), liquid (b) and total (c) extinctions from VarPy-mix retrievals (curtain) and *in situ* probes (dots) regarding the latitude and the height. The panels (d) to (f) share the same ordinate axis and represent the ice (d), liquid (e) and total (f) extinctions from VarPy-mix retrievals and *in situ* probes regarding the latitude. The error bars of *in situ* measurements (uncertainties from Table 6) are displayed in panels (d) to (f). The yellow and purple shading represents the latitude range where mixed-phase retrievals are compared with *in situ*.

[Figure]

FIGURE 5 – As Fig. 4 for IWC, LWC and TWC.

**Références**

GAYET, Jean-François, Frédérique AURIOL, Andreas MINIKIN, Johan STRÖM, Marco SEIFERT, Radovan KREJCI, Andreas PETZOLD, Guy FEBVRE et Ulrich SCHUMANN (2002). "Quantitative measurement of the microphysical and optical properties of cirrus clouds with four different in situ probes : Evidence of small ice crystals". en. In : *Geophysical Research Letters* 29.24. _eprint : https ://onlinelibrary.wiley.com/doi/pdf/10.1029/2001GL014342, p. XXX-XXX. ISSN : 1944-8007. DOI : 10.1029/2001GL014342. URL : https://onlinelibrary.wiley.com/doi/abs/10.1029/2001GL014342 (visité le 18/11/2023) (cf. p. 5).

JOURDAN, Olivier, Sergey OSHCHEPKOV, Valery SHCHERBAKOV, Jean-Francois GAYET et Harumi ISAKA (sept. 2003). "Assessment of cloud optical parameters in the solar region : Retrievals from airborne measurements of scattering phase functions". en. In : *Journal of Geophysical Research : Atmospheres* 108.D18, 2003JD003493. ISSN : 0148-0227. DOI : 10.1029/2003JD003493. URL : https://agupubs.onlinelibrary.wiley.com/doi/10.1029/2003JD003493 (visité le 20/11/2023) (cf. p. 5).

JOURDAN, Olivier, Guillaume MIOCHE, Timothy J. GARRETT, Alfons SCHWARZENBÖCK, Jérôme VIDOT, Yu XIE, Valery SHCHERBAKOV, Ping YANG et Jean-François GAYET (2010). "Coupling of the microphysical and optical properties of an Arctic nimbostratus cloud during the ASTAR 2004 experiment : Implications for light-scattering modeling". en. In : *Journal of Geophysical Research : Atmospheres* 115.D23. _eprint : https ://onlinelibrary.wiley.com/doi/pdf/10.1029/2010JD014016. ISSN : 2156-2202. DOI : 10.1029/2010JD014016. URL : https://onlinelibrary.wiley.com/doi/abs/10.1029/2010JD014016 (visité le 07/11/2023) (cf. p. 5).

FEBVRE, G., J.-F. GAYET, V. SHCHERBAKOV, C. GOURBEYRE et O. JOURDAN (oct. 2012). "Some effects of ice crystals on the FSSP measurements in mixed phase clouds". en. In : *Atmospheric Chemistry and Physics* 12.19, p. 8963-8977. ISSN : 1680-7324. DOI : 10.5194/acp-12-8963-2012. URL : https://acp.copernicus.org/articles/12/8963/2012/ (visité le 03/03/2024) (cf. p. 3).

LEINONEN, Jussi et Wanda SZYRMER (2015). "Radar signatures of snowflake riming : A modeling study". en. In : *Earth and Space Science* 2.8. _eprint : https ://onlinelibrary.wiley.com/doi/pdf/10.1002/2015EA000102, p. 346-358. ISSN : 2333-5084. DOI : 10.1002/2015EA000102. URL : https://onlinelibrary.wiley.com/doi/abs/10.1002/2015EA000102 (visité le 03/03/2024) (cf. p. 4).

COSTA, Anja, Jessica MEYER, Armin AFCHINE, Anna LUEBKE, Gebhard GÜNTHER, James R. DORSEY, Martin W. GALLAGHER, Andre EHRLICH, Manfred WENDISCH, Darrel BAUMGARDNER, Heike WEX et Martina KRÄMER (oct. 2017). "Classification of Arctic, midlatitude and tropical clouds in the

mixed-phase temperature regime". en. In : *Atmospheric Chemistry and Physics* 17.19, p. 12219-12238. ISSN : 1680-7324. DOI : `10.5194/acp-17-12219-2017`. URL : `https://acp.copernicus.org/articles/17/12219/2017/` (visité le 03/03/2024) (cf. p. 3, 7).

**Lidar-radar synergistic method to retrieve ice, supercooled and mixed-phase clouds properties**
**List of changes**

Following discussions with the referees, we made a number of modifications and improvements to the manuscript. This document summarizes all the changes that have been added to the revised version of the manuscript. The first section covers changes to figures and tables. The second section is dedicated to the changes made to the abstract and to the text of the different sections of the manuscript.

**Changes to figures and tables**

We present here the changes relating to figures and tables, in the order in which they appear in the document :

- [p. 5] Figure 1 has been corrected :
  — the arrow between boxed 6 and 7 has been removed.
  — the numbering of boxes 6 and 7 has been switched.
  — box 9 composition is more detailed (corresponding to box 1 VarPy-ice/VarPy-mix).
- [p. 16] Table 4 : the term "crystals" has been replaced by "ice particles".
- [p. 16] Table 5 : the term "crystals" has been replaced by "ice particles".
- [p. 20] Table 6 :
  — two rows has been added with information on number concentration and asymmetry parameter
  — the caption has been updated by replacing "extinction and water contents" by "cloud properties derived"
- [p. 21] Figure 4 has been improved :
  — panels has been switched : (a) with (b) and (d) with (e).
  — y axis of panels (d), (e) and (f) are now the same.
- [p. 22] Figure 5 has been improved :
  — panels has been switched : (a) with (b) and (d) with (e).
  — y axis of panels (d), (e) and (f) are now the same.
- [pp. 23 and 24] Figure 6 has been replaced by two new Figures (6 and 7)
- [p. 24] Table 7 :
  — the tenth line value and unit have been changed to $2.01 \times 10^{-2} \mathrm{cm}^{-3}$ instead of $2.01 \times 10^{4} \mathrm{m}^{-3}$.
  — the eleventh line value has been corrected and the unit have been changed : $3.73 \times 10^{1} \mathrm{cm}^{-3}$ instead of $1.99 \times 10^{7} \mathrm{m}^{-3}$.
  — a twelfth line has been added to include the mean value for total number concentration $N_{\mathrm{tot}}$ equals to $1.99 \times 10^{1} \mathrm{cm}^{-3}$.
- [p. 26] Table 8 :
  — information on the effective radii ($r_{e,\mathrm{ice}}$ and $r_{e,\mathrm{liq}}$) and the number concentrations ($N_{\mathrm{ice}}$, $N_{\mathrm{liq}}$ and $N_{\mathrm{tot}}$) have been added, implying five additional rows at the end of the table
  — two columns have been added in the middle of the table with information on the "Mean values for VarPy-mix selected gates" and "Mean values for *in situ*"

**Changes to text**

We present here the changes made to the text. First, for the abstract :

- [l. 7] : We have added ", called VarPy-mix" at the end of the sentence.
- [l. 7] : We have replaced "method" by "new approach".
- [ll. 11 and 12] : We have replaced the term "crystals" by "particles".
- [l. 11] : We have removed "liquid" from "supercooled liquid droplets".
- [ll. 21-22] : We have removed the link word "and" to add "effective radii and number concentrations".
- [ll. 24-28] : We have replaced "It is also important to note that the temporal and spatial collocations are not always optimal, that the sensibility of remote sensing and *in situ* are not the same and that *in situ* measurements uncertainties are between 25 % and 60 %." by "It is also important to note that temporal and spatial collocations are not perfect, with a maximum spatial shift of 1.68 km and a maximum temporal shift about ten minutes between the two platforms. In addition, the sensitivity of remote sensing and *in situ* are not the same and *in situ* measurements uncertainties are between 25 % and 60 %." to add information on the temporal and spacial shifts. We have therefore split the sentence in two.

Finally, for the sections of the manuscript :

- [ll. 35, 40, 62, 81, 183, 197, 200, 222, 254, 255, 301, 336, 340, 436, 437, 502 and 503] : We replaced the term "crystals" by "particles".
- [l. 47] : We have corrected "localisation" to "localization".
- [ll. 53-55] : We have added some information related to the term "attenuated backscatter" and slip the original sentence in two.
- [ll. 57-59] : We have improved the sentence and split it in two.
- [ll. 61 and 62] : We have corrected the tense used and replaced "will give" by "gives" and "is".
- [ll. 62-64] : We have improved the sentence and split it in two.
- [ll. 74-76] : We have split the sentence in two to lighten the text and improve comprehension.
- [l. 78] : We have replaced "The method that we propose" by "Our method".
- [l. 87] : We have replaced "The cloud phases that are processed are also" by "In addition, the processed cloud phases are".
- [l. 88] : We have replaced "Then" by "Next".
- [l. 95] : We have replaced "apparent" by "attenuated".
- [ll. 97-103] : We have improved this part by added "$\mathbf{Y} = f(\mathbf{X}) + \epsilon$" as an equation and split the original sentence in three.
- [ll. 107-108] : We have split the sentence in two to lighten the text and improve comprehension.
- [l. 167] : We have corrected the sentence by replacing "both LUT are used" by "both LUT can be used" and adding "and one must be selected beforehand".
- [l. 185] : We have corrected "is" to "are".
- [l. 208] : We have improved the sentence by adding "This relation is" and replacing "used for" by "to calculate".
- [ll. 208-209] : We have replaced "the benefits of the old scheme" by "the old scheme benefits".
- [ll. 236-238 and 246-247] : We have split the sentences in two to lighten the text and improve comprehension.
- [ll. 265-276] : We have improved this part by organizing information in list form to improve readability.
- [l. 301] : We have replaced "therefore" by "meaning that".
- [l. 302] : We have replaced "input of the algorithm " by "algorithm input ".
- [l. 308] : We have corrected "categorisation" to "categorization".
- [ll. 313, 317, 337 and 338] : We have replaced the term "crystals" by "ice particles".
- [l. 315] : We have corrected "formed" to "form".

- [ll. 326-327] : We have replaced "there is no radar signal and a strong lidar backscatter" to "the radar does not detect particles (no radar signal) and the lidar backscatter is strong".
- [ll. 341-356] : We have added one subsection "2.4 Summary of the methodology" that summarizes the five main key points of VarPy-mix.
- [ll. 391-421] : We have improved this part by :
  — discussing the detection of secondary ice particles by the FSSP. We assume here that the presence of small ice crystals does not significantly impact the results, since the asymmetry parameter is greater than 0.84, indicating the almost exclusive presence of water droplets.
  — adding the mass-size relationship used to derived the ice water content from the CPI ($m = 0.033 \times D^{1.94}$).
  — explaining how we derive the ice and liquid effective radii for *in situ* probes.
  — adding information about effective radii and number concentration between line 417 and 421 (i.e. reference to Fig. 6 and 7).
- [ll. 423-424] : We have corrected the sentence by replacing "The HC mass-size relationship is used in this study to retrieve ice properties." by " In this study, we chose to retrieve ice properties with the HC LUT".
- [ll. 428 and 429] : We have replaced "ice and liquid" by "liquid and ice" for consistency.
- [l. 430] : We have replaced "IWC and LWC" by "LWC and IWC" for consistency.
- [ll. 430-431] : We have replaced "and Fig. 6 (a) to (d) show $N_{\mathrm{ice}}$, $N_{\mathrm{liq}}$, $r_{e,\mathrm{ice}}$ and $r_{e,\mathrm{liq}}$." by "Fig. 6 (a) and (b) show $r_{e,\mathrm{ice}}$ and $r_{e,\mathrm{liq}}$ and Fig. 7 (a) and (b) show $N_{\mathrm{ice}}$ and $N_{\mathrm{liq}}$" for consistency with our figure modifications.
- [ll. 434-435] : We have corrected "With these values and a global view with the curtains, it is possible to observe trends for each variable. These values allow us to observe trends for each variable" with "These values allow us to observe trends for each variable".
- [ll.] : We have corrected "Both extinctions and water contents" to "All retrieved variables " for consistency.
- [ll. 466-476] : We have added one paragraph about the effective radii and number concentrations comparison. We discuss the difference between VarPy-mix retrievals and *in situ* measurements.
- [ll. 525-526] : We have replaced "the extinctions and the water contents from" by " the cloud microphysical properties retrieved with" for consistency.
- [ll. 537-539] : We have added two sentences to explain that there are uncertainties in the comparison between VarPy-mix and *in situ* due to the different particle size distribution used.
- [ll. 544-545] : We have added "hat took place in August 2022 near the Svalbard archipelago" to provide some context about the RALI-THINICE campaign.
- [ll. 547-553] : We have added information on the HALO-(AC)[3] campaign as well as the Overview Paper related to the campaign. In addition, we have added the website related to both field campaign.